# Research using population-based administration data integrated with longitudinal data in child protection settings: A systematic review

**Fadzai Chikwava**[1] *, **Reinie Cordier**[1,2], **Anna Ferrante**[3], **Melissa O'Donnell**[4,5], **Renée Speyer**[1,6,7], **Lauren Parsons**[1]

1 School of Allied Health, Curtin University, Perth, Western Australia, Australia, 2 Department of Social Work, Education and Community Wellbeing, Northumbria University, Newcastle, United Kingdom, 3 School of Public Health, Curtin University, Perth, Western Australia, Australia, 4 Telethon Kids Institute, University of Western Australia, Nedlands, Western Australia, Australia, 5 Australian Centre for Child Protection, University of South Australia, Adelaide, South Australia, Australia, 6 Department of Special Needs Education, University of Oslo, Oslo, Norway, 7 Department of Otorhinolaryngology and Head and Neck Surgery, Leiden University Medical Centre, Leiden, The Netherlands

* Fadzai.Chikwava@postgrad.curtin.edu.au

**Data Availability Statement:** All relevant data are within the manuscript and its Supporting Information files.

## Abstract

### Introduction

Over the past decade there has been a marked growth in the use of linked population administrative data for child protection research. This is the first systematic review of studies to report on research design and statistical methods used where population-based administrative data is integrated with longitudinal data in child protection settings.

### Methods

The systematic review was conducted according to Preferred Reporting Items for Systematic Review and Meta-Analyses (PRISMA) statement. The electronic databases Medline (Ovid), PsycINFO, Embase, ERIC, and CINAHL were systematically searched in November 2019 to identify all the relevant studies. The protocol for this review was registered and published with Open Science Framework (Registration DOI: **10.17605/OSF.IO/96PX8**)

### Results

The review identified 30 studies reporting on child maltreatment, mental health, drug and alcohol abuse and education. The quality of almost all studies was strong, however the studies rated poorly on the reporting of data linkage methods. The statistical analysis methods described failed to take into account mediating factors which may have an indirect effect on the outcomes of interest and there was lack of utilisation of multi-level analysis.

**Funding:** The principal author (FC) is in receipt of the Australian Government Research Training Program (RTP)(https://www.education.gov.au/research-training-program) Scholarship and The Australian Housing and Urban Research Institute (AHURI)(https://www.ahuri.edu.au/) Scholarship. The funders had no role in study design, data collection and analysis, decision to publish, or preparation of the manuscript.

**Competing interests:** The authors have declared that no competing interests exist.

## Conclusion

We recommend reporting of data linkage processes through following recommended and standardised data linkage processes, which can be achieved through greater co-ordination among data providers and researchers.

## Introduction

Population-based administrative data is routinely collected by organisations to deliver services and to monitor, evaluate and improve upon those same services [1]. Some examples of the types of data include administrative health data, disease registries, primary care databases, electronic health records, population registries and birth and death registries [2]. The data may be linked within a single service sector, such as health, or with surveys and across sectors such as education, child protection and corrective services [1, 3, 4]. Bringing together data from various administrative data sources provides a rich repository of data that can be used for research purposes. The linked data enables researchers to study risk and protective factors and to examine outcomes from various databases brought together [5, 6]. The trend of using administrative data for research purposes has increased exponentially [7–13]. To date, there has not been a systematic review that has focussed on methods of analysis of integrated population-based administrative data with longitudinal data in child protection settings.

Population-based administrative data is invaluable in research as it offers complete coverage of a given population which overcomes the imprecision associated with sampling errors [14]. It offers superior statistical power and precision to determine associations between rare exposures and outcomes, and using these samples as sampling frames for subsequent surveys [1, 15–18]. Administrative data is useful when studying causes of complex diseases and conditions as well as assessing outcomes of clinical or therapeutic interventions [17, 19–21]. Use of multiple linked administrative data allows researchers to explore comorbidity and variability in outcomes within target populations and compare these between specific clinical population groups and against outcomes in the general population [22–25]. As the purpose of this systematic review is related to child protection settings, it will be used as an example to elucidate the benefits and limitations of using population-based administrative integrated with longitudinal data in research.

Population-based administrative data allows the study of outcomes among cohorts of hard to reach or high-risk populations such as those in the juvenile justice system, and those involved with the child protection system [15, 26, 27]. For example, child protection administrative data allow longitudinal examination of population-level patterns and trends in child maltreatment and complex multi-level analysis, particularly where the data is linked to individuals who are related [27–31]. The data allow the determination of cumulative incidence of risk and protective factors among various population subgroups with different levels of child protection involvement [22, 32, 33]. Therefore the data allows researchers to trace various trajectories of specific cohorts from birth to adulthood [34].

Use of child protection administrative data in research reduces the burden on individuals to disclose sensitive or traumatic experiences and also reduces the risk of recall bias, social desirability and stigma, which may occur, for instance, in retrospective self-report of child maltreatment [4, 27]. Administrative data is less prone to selection bias since the data includes the entire population served by the Child Protection Agency. Such data is also used to evaluate the frequency of use, effectiveness and costs of services across populations and over time [35].

Further, using administrative data is more cost-effective and efficient in that data is readily available when needed [36] and one can avoid the cost and burden associated with face to face data collection.

Despite all the advantages of using population-based administrative date, there are some limitations to using and accessing administrative data. Key variables of interest to researchers are often not recorded since administrative data are primarily collected for the delivery of programs and services [14]. The data may be subjected to biases, such as under-reporting of the incidence of child maltreatment in child protection research or lack of availability of data for some respondents, particularly difficulty in reaching vulnerable groups [1]. In addition, the type of data being collected routinely may lack the depth of information required to answer important research questions [27]. Another important limitation of administrative data is that individual-level socio-economic status (SES) parameters are often not available [37].

Linked administrative data may be subject to linkage error when some records that should be matched or able to be linked were not linked (missed matches) or records were linked incorrectly (false matches), which could lead to biased estimates of association [7, 38]. There are also data access challenges, such as delays in getting approvals to link datasets, especially getting access to cross-jurisdictional linked datasets [26]. There may be restrictions placed by data custodians on who may access linked data, thereby limiting the ability of researchers to access all the data they may need [1]. Despite the above limitations of using population-based administrative data alone, there are advantages of linking population based administrative data to longitudinal data.

The benefits of conducting longitudinal research in child protection settings are well documented, as this type of research allows researchers to analyse trends, changes in early exposures, risks, behaviours and outcomes over a long period of time [18, 39]. Longitudinal studies are also powerful in that they overcome common issues around temporal associations and causal risk factors for outcomes of child abuse and neglect [5]. Longitudinal studies also allow researchers to update certain information about participants, such as socio-demographic characteristics, and also obtain in-depth information about certain topics and service involvement, which otherwise could not be collected from administrative data alone [18, 40].

Despite the notable benefits of conducting longitudinal studies, they are known to be notoriously expensive as they involve several waves of data collection, and could run for several years before the outcomes of a study are determined [37]. It may be difficult to obtain sufficient numbers of eligible participants, particularly when recruiting hard to reach populations and access to children in out-of-home care is generally tightly controlled, resulting in low response rates [41]. Longitudinal data are also subject to different biases such as under-reporting, recall errors and high attrition rates [18], resulting in reporting of biased estimates if the biases are not appropriately accounted for in the analysis. A systematic review conducted by Farzanfar, Abumuamar [42] highlighted the potential for bias and on the reporting of longitudinal studies. Another review by Karahalios, Baglietto [43], found that 56% of studies had a high risk of bias with regards to attrition. Longitudinal studies also place a high burden on respondents due to frequent contact.

Combining population-based administrative data with longitudinal data has several advantages. For example, linking child protection administrative data to longitudinal data allows use of retrospective administrative data on prenatal or early childhood experiences to determine a trajectory of long term adult outcomes which can be measured from longitudinal data [44–47]. Young people who have had child protection contact are known to have worse outcomes than young people in the general population [48, 49]. Thus, integrating longitudinal data and administrative data enables comparison of outcomes using population level data. Other benefits of linking longitudinal data with administrative data include the following: i) cross-

validation of self-reported information from longitudinal surveys with administrative data [26, 38, 50, 51]; ii) reducing data incompleteness and biases inherent in longitudinal data as reported earlier [40, 52, 53]; and iii) overcoming high attrition rates common in longitudinal data [52, 54, 55]. In summary, combining these two data sources increases the usability and possible applications of the data.

Using population-based administrative data integrated with longitudinal data has its own limitations. One of the challenges is the introduction of bias by linking data only where consent has been provided by respondents [1, 56]. Further, the linkage may be of poor quality and the data from administrative records may not exist or be incomplete for many longitudinal participants [1].

A wide variety of factors affect the accuracy of reported results in child protection settings. These include the reference population, data source, sampling strategy, sample size and analytical factors [41, 57]. While data integration offers unique advantages, it is important to consider various techniques and methods of analysis to report study outcomes and to correct for biases which may be introduced by bringing together data from various sources. When modelling outcomes using administrative data integrated with longitudinal data it is important to consider time between occurrences of events (survival analysis), all possible confounders, and mediating and moderating factors. These may include early childhood experiences, pre-natal and parental risk factors, socio-demographic and environmental factors [58]. Failure to account for these factors may lead to biased estimates and false inference. Sensitivity analysis may be conducted to investigate the extent to which some changes or modifications in the confounding variables may have an effect on reported outcomes. For example, multiple regression models may be constructed involving child maltreatment notifications as a risk factor compared to modelling substantiated maltreatment on outcomes [45, 59].

Some of the considerations that need to be taken into account when analysing these datasets involve methods of dealing with biases in the datasets. Missing data can lead to biased estimates of regression parameters when the probability of missingness is associated with outcomes. Different strategies are used to handle missing data in statistical analyses, such as: i) imputation of missing data, [60, 61]; ii) using maximum likelihood estimation methods to model data from subjects who drop out of the study compared to those who complete the longitudinal study; and iii) weighting the available data using non-response methods to account for missing data [62, 63]. Some concurrence or agreement tests may need to be conducted to determine validity of responses from either data sources [64–66].

Some studies have demonstrated that longitudinal data analysis should account for possible within-subject correlation and different covariance structures of episodes of various disease outcomes over time. Some of the analytical methods used for this include generalized estimating equations (GEE) and mixed-effects models [67–71].

Previous reviews have focused on measurement of the diagnosis of diseases or outcomes, including administrative data characteristics and strengths and limitations of the two data sources [17, 72–74]. A systematic review conducted by Tew, Dalziel [26] focussed on the use of linked hospital data for research in Australia, thereby limiting the generalisability of the findings. Young and Flack [13] conducted a review that reported on recent trends of using linked data. Even though this paper used systematic search strategy, it was not published as a systematic review. In addition, the study highlighted areas where linked data is commonly used, particularly in cross-sectorial linked data and areas where its use could be improved, however it did not mention use of longitudinal data to enhance reporting of outcomes. A systematic review conducted by Andrade, Elphinston [75], highlights the need for future research to focus on collecting better measures for outcomes data and linking data to multiple

administrative databases. A systematic review conducted by da Silva, Coeli [76] examined the issue of consent for data linkage, which is one of the sources of bias in using linked data.

Selecting appropriate statistical analysis of administrative data integrated with longitudinal data can improve the reporting of risk and protective factors related to child protection outcomes. This can be achieved through careful selection of variables and optimal use of the data extracted from the administrative and longitudinal data. The over-arching aim of this review is to provide a synthesis of the different methods of analysis used when administrative data is integrated with longitudinal data and make recommendations about approaches to enhance research findings thereby minimising risk of bias and other limitations. Specifically, the following objectives will be investigated: i) to describe the study designs and methods used in reporting linked administrative data when combined with longitudinal data in child protection settings; and ii) to identify statistical methods, gaps and opportunities in the analysis of administrative data integrated with longitudinal data in child protection settings.

Although research on combining administrative data integrated with longitudinal data in child protection research is available, to the best of our knowledge, no systematic reviews have reported on the statistical methods used when the two data sources are combined. This systematic review is an essential step towards informing policy, practice and future research directions in methodological aspects of using administrative data integrated with longitudinal data in child protection settings.

## Methods

The systematic review was conducted according to Preferred Reporting Items for Systematic Review and Meta-Analyses (PRISMA) statement [77] which outlines minimum standards for reporting systematic reviews and meta-analysis. A completed PRISMA checklist is provided in S1 Table. The protocol for this review was registered and published with Open Science Framework (Registration DOI: **10.17605/OSF.IO/96PX8**).

### Eligibility criteria

To be included in this review, peer reviewed studies needed to have at least one administrative database integrated with a longitudinal data. Selected studies were limited to studies involving child protection settings and published in English only. Studies involving systematic reviews or meta-analysis were excluded. In addition, anecdotes, reviews, book chapters, letters to the editor, editorials and conference abstracts were excluded. Studies had to meet all eligibility criteria to be included in the review.

### Information sources and search strategy

The electronic databases Medline (Ovid), PsycINFO, Embase, ERIC, and CINAHL were systematically searched in November 2019 to identify all the relevant studies. In line with the objective of this review, terms were identified in electronic databases that are related to the following three concepts: i) data source (administrative data or population based data); ii) study design (longitudinal study or cohort study or prospective study); and iii) setting (child protection). Searches were conducted using free-text in all databases because we had too few relevant subject headings for our purposes. In addition, websites that provide a publication repository for studies involving linked data, such as the Population Health Research Network, were searched. The reference list of included studies was manually searched to find additional relevant studies. A full search strategy for all databases is shown in S2 Table.

## Study selection

Screening of titles and abstracts of the retrieved studies was conducted between December 2019 and March 2020. The first author screened all titles and abstracts while the second reviewer (LP) independently screened a random selection of 40% of studies to identify the candidate studies for the full text review. The reviewers graded each abstract as eligible, possibly eligible or not eligible (using the inclusion and exclusion criteria defined above). Both reviewers independently screened 100% of full-text studies. Any disagreements about eligibility of full-text studies were settled by discussing the differences in the assessment and reaching a consensus on which studies to include. Five studies were used to pilot the screening criteria, and data extraction process, which were modified after consultation between researchers. Inter-rater reliability using weighted Kappa between the two independent reviewers was established for the abstract selection and quality appraisal of included studies. The weighted Kappa measures the degree of disagreement between the two raters; the greater the disagreement the higher the weight.

## Methodological quality

Since there is no standard criteria for assessing the quality of study designs involving integration of population-based administrative data and longitudinal data, a combination of three critical appraisal methods for assessing the methodological quality of studies was utilised. The critical appraisal methods were the "Qualsyst" critical appraisal tool by Kmet et al. [78] (henceforth referred to as kmet checklist), the Guidance for Information about Linking Data sets (GUILD) [7], which focus on the methodological process of linking data, and the Reporting of studies Conducted using Observational Routinely-collected health Data (RECORD) [79].

The Kmet checklist has 14 items that use a 3-point ordinal scale (0 = no, 1 = partial, 2 = yes) of which three items were not applicable to our study design. The checklist items assess the study design, description of participants' characteristics, appropriateness of sampling strategy and sample size, robustness of outcome and exposure variables, analytical methods, estimates of variance, control for confounding and whether conclusions drawn reflect results reported. A Qualsyst score of > 80% was interpreted as strong quality, 60–79% as good quality, 50–59% as adequate quality, and < 50% as poor methodological quality.

The GUILD statement has three broad domains with items within each domain that focus on the data source population and linkability of the dataset, data linkage process and quality of data linkage including accounting for linkage error. The RECORD statement, an extension from the STROBE guidelines, consists of a checklist of 13 items related to the title, abstract, introduction, methods, results, and discussion section of studies and other items relating to routinely collected health data [79]. Three items were selected from the RECORD checklist as they were the only items that did not overlap with the GUILD items; these items were combined with the GUILD statements. Due to the absence of a standard scoring system for the GUILD and RECORD statements, a similar scoring method to Kmet was used. Prior to conducting the quality appraisal, the two reviewers (FC and LP) met to discuss the scoring method for these guidelines.

The second reviewer conducted quality assessment (using Kmet, GUILD and RECORD statements) on a random selection of 40% of the included studies. Any differences in ratings from the two reviewers were settled by discussing the differences in the assessment and reaching a consensus on the final score for each of the quality appraisal methods. The differences for Kmet were defined as any difference in the rating from one category to the next (e.g., when a study was rated as good quality (60–79%) by one reviewer, while the same study is rated as poor quality (<50%) by the other reviewer). However, because most studies received poor

GUILD and RECORD ratings, discussions on agreement between scores were conducted for GUILD and RECORD ratings with more than 15% difference for each study.

## Data collection process

Comprehensive data extraction forms were developed to extract relevant data from the included studies under the following four headings: study characteristics, administrative data, longitudinal data and statistical methods. The included studies were heterogeneous in terms of study design and quality, therefore a narrative synthesis of the findings of the included studies was conducted.

# Results

## Study selection

A total of 1,123 studies were retrieved from the electronic database search and eight from other sources. Out of these, a total of 698 studies remained after duplicates were removed. A total of 664 records did not meet the inclusion criteria, resulting in 34 full-text studies which were assessed for eligibility. The final number of studies that met the inclusion criteria and were included in data synthesis were 30 and of these 10 were identified by manually scrutinising the references of the eligible studies. **Fig 1** below shows a flowchart of the search and selection process of the included studies.

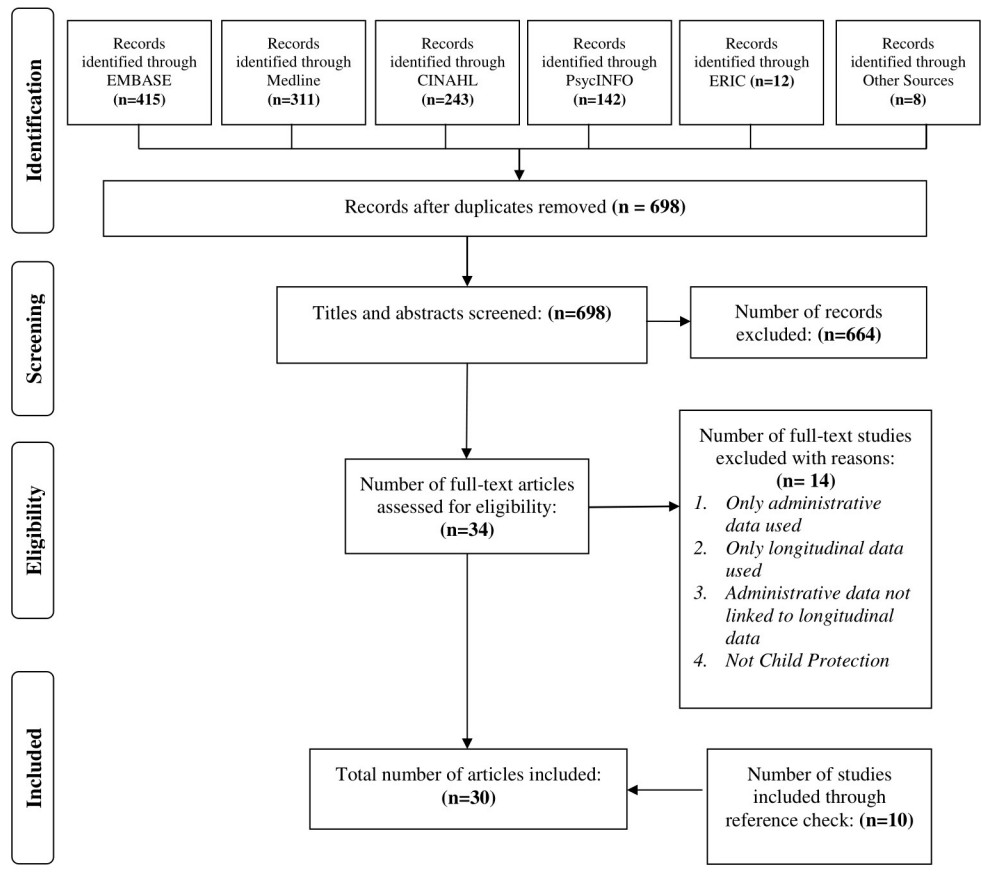

**Fig 1. PRISMA flow diagram.**

## Characteristics of included studies

The studies were conducted in a variety of countries with Australia having the highest number of publications (50%), followed by the USA (20%) and the United Kingdom (17%). While all studies were conducted in child protection settings, only a few were specific to out-of-home care settings (20%). The outcomes investigated were varied; the most common outcomes were child maltreatment (30%), mental health (20%), drugs and alcohol abuse (20%), education (17%), domestic violence (7%), and health insurance (7%). Table 1 below shows a summary of all included studies, and Table 2 has more detailed information for each study.

Almost all studies were birth cohorts and they each measured different variables at different points in time. In the majority of studies, baseline data consisted of prenatal or postnatal data as reported by the mothers, while outcome data were obtained during follow-up waves. Six major longitudinal studies were reported from the publications, the main one being the Mater-University Study of pregnancy (MUSP) which was conducted in Queensland, Australia from 1981–2004 [58, 80–82]. While these studies had multiple follow-up waves, the authors mostly reported on the baseline wave and one follow up wave. The duration of follow up from the baseline to the last wave ranged from 3 to 21 years. Each longitudinal study had multiple publications demonstrating that a range of exposures and outcomes can be investigated in linked child protection datasets. There was an almost equal number of males and females reported in 70% of studies, while the gender split was unknown in 9 studies.

**Table 1. Characteristics of the study population.**

| Characteristic | | N | % |
|---|---|---|---|
| Country | Australia | 15 | 50% |
| | USA | 6 | 20% |
| | UK | 5 | 17% |
| | Denmark | 2 | 7% |
| | Sweden | 2 | 7% |
| Research Area | Child Protection | 9 | 30% |
| | Drugs & Alcohol | 6 | 20% |
| | Mental Health | 6 | 20% |
| | Education | 5 | 17% |
| | Domestic Violence | 2 | 7% |
| | Health Insurance | 2 | 7% |
| Population group | Child protection Contact | 24 | 80% |
| | Out-of-home care | 6 | 20% |
| Linkage Method | Deterministic | 17 | 57% |
| | Probabilistic | 2 | 7% |
| | Deterministic & Probabilistic | 2 | 7% |
| | Not reported | 9 | 30% |
| Admin datasets | 1 | 25 | 83% |
| | >1 | 5 | 17% |
| Name of Longitudinal Study | The Mater-University Study of pregnancy (MUSP) | 14 | 47% |
| | Alaska Pregnancy Risk Assessment Monitoring System (PRAMS) | 5 | 17% |
| | The Avon Longitudinal Study of Parents and Children (ALSPAC) | 5 | 17% |
| | Danish longitudinal survey of children (DALSC) | 5 | 17% |
| | Swedish longitudinal Evaluation Through Follow-up (ETF) project | 2 | 7% |
| | National Survey of Child and Adolescent Well-Being (NSCAW) | 2 | 7% |

**Table 2. Characteristics of included studies.**

| Author (Year) | Country | Aims/ Objectives | Research Area | Child Protection Contact (CPC) vs. OHC | Administrative Data Source | Number of administrative datasets (Deterministic/ Probabilistic Linkage) | Linkage Quality (Yes/ No) |
|---|---|---|---|---|---|---|---|
| Egulend et al. (2009) | Denmark | To identify problems among children in foster and residential care compared to in home care children, and to all non-welfare children of the same age, and to analyse factors associated with mental health problems in children in out-of-home care | Mental Health | OHC | 1.National Health Register; 2.Psychiatric Research Register 3.Child Protection Register | 2 (Deterministic) | No |
| Hansson et al. (2018) | Sweden | To describe and discuss differences between children placed in OHC and non-OHC children in the Swedish compulsory school, with respect to special needs education, school mobility and academic achievement. | Education | OHC | Statistics Sweden | 1 (NR) | No |
| Kisely et al. (2019) | Australia | To examine whether notified and/or substantiated child maltreatment is associated with the prevalence and persistence of smoking in early adulthood | Drugs & Alcohol | CPC | Queensland Department of Families, Youth and Community Care (DFYCC) | 1 (Deterministic) | No |
| Kisely et al. (2018) | Australia | To examine, using a prospective record-linkage analysis, whether substantiated child maltreatment is associated with adverse psychological outcomes in early adulthood. | Mental Health | CPC | Queensland Department of Families, Youth and Community Care (DFYCC) | 1 (Deterministic) | No |
| Kisely et al. (2019) | Australia | To study the association of different types of child maltreatment with alcohol use disorders at 21 years of age | Drugs & Alcohol | CPC | Queensland Department of Families, Youth and Community Care (DFYCC) | 1 (Deterministic) | No |
| Olsen et al. (2018) | Denmark | To investigate the association for children in OHC and non-OHC peers between school change in lower secondary school and two educational outcomes: (1) self-perceived academic abilities at age 15 and (2) staying-on rates in upper secondary school at age 18 | Education | OHC | Danish Register Data | 1 (Deterministic) | No |
| Parrish et al. (2016) | USA | To determine the predictive relationship between a maternal pre-birth self-reported history of intimate partner violence (IPV) and any post-birth reported allegation to Child Protective Services (CPS) by age 2 | Domestic violence | CPC | Alaska's Child Protective Services Agency Register | 1 (Probabilistic) | No |
| Parrish et al. (2017) | USA | A description of the creation of the (ALCANLink) project and the benefit of the ALCANLink methodology by documenting the bias in incidence and hazard ratios that can arise in birth cohort linkage studies due to incomplete data linkages, non-linkage assumptions, and single source outcome ascertainment | Child protection | CPC | 1. Vital records; 2. Child death review; 3. Alaska Permanent Fund Dividend (PFD) records | 3 (Deterministic & Probabilistic) | Yes |
| Raghavan et al. (2017) | USA | To quantify the magnitude of non-ascertainment bias, develop a profile of children who are at greatest risk for non-ascertainment, | Health insurance | OHC | 1.Medicaid Analytic eXtract (MAX) Research Data Assistance Centre; 2.Child Welfare Agency | 1 (Deterministic) | Yes |
| Sidebotham et al. (2000) | UK | A study of patterns of child abuse and factors that may affect risk in a pre-school population | Child protection | CPC | Avon Social Services Child Protection Register | 1 (NR) | No |
| Sidebotham et al. (2003) | UK | To determine characteristics of children that may predispose to maltreatment. | Child protection | CPC | Avon Social Services Child Protection Register | 1 (NR) | No |
| Sidebotham et al. (2006) | UK | to analyse the multiple factors affecting risk of abuse in young children within a comprehensive theoretical framework | Child protection | CPC | Avon Social Services Child Protection Register | 1 (NR) | No |

*(Continued)*

**Table 2.** (Continued)

| Author (Year) | Country | Aims/ Objectives | Research Area | Child Protection Contact (CPC) vs. OHC | Administrative Data Source | Number of administrative datasets (Deterministic/ Probabilistic Linkage) | Linkage Quality (Yes/ No) |
|---|---|---|---|---|---|---|---|
| **Sidebotham et al. (2002)** | UK | To determine risk factors for child maltreatment within the socio-economic environment of a contemporary UK child population | Child protection | CPC | Avon Social Services Child Protection Register | 1 (NR) | No |
| **Teyhan et al. (2019)** | UK | To use record linkage of birth cohort and administrative data to study educational outcomes of children who are looked-after (in public care) and in need (social services involvement), and examine the role of early life factors. | Education | OHC | 1. Children Looked-After (CLA) Data Return; 2. Children in Need (CIN) Census; 3. National Pupil Database | 3 (NR) | No |
| **Austin et al. (2019)** | USA | Identify longitudinal trajectory classes of CPS contact among Alaska Native (AN/AI) and non-Native (NN) children and examine preconception and prenatal risk factors associated with identified classes | Child protection | CPC | 1. Alaska Office of Children's Services (OCS); 2. Alaska Child Death Review; 3. Death certificate files; 4. Alaska Dept. of Revenue | 4 (NR) | No |
| **Austin et al. (2018)** | USA | To use multiple novel data sources and time-to event analysis to examine preconception and prenatal predictors of time to first contact with CPS among a representative sample of Alaska children. | Child protection | CPC | 1. Alaska Office of Children's Services (OCS); 2. Alaska Child Death Review; 3. Death certificate files; 4. Alaska Dept. of Revenue 5. Geographic census classification data 6. Alaska Birth Defects Registry | 6 (NR) | No |
| **Hansson et al. (2020)** | Sweden | To investigate the effects of school mobility on academic achievements for OHC-children as well as for NOHC-children. | Education | OHC | Statistics Sweden: Child Welfare Register | 1 (NR) | No |
| **Abajobir et al. (2017)** | Australia | Examine the association between different types of substantiated child maltreatment and self-reported psychotic experiences as measured by the Young Adult Self-Report (YASR) items and the Peter's Delusions Inventory (PDI) using data from a large population-based birth cohort study. | Mental Health | CPC | Queensland Department of Families, Youth and Community Care (DFYCC) | 1 (Deterministic) | No |
| **Abajobir et al. (2017)** | Australia | Examine the effect on QoL of multiple forms of substantiated child maltreatment controlling for selected potential confounders and/ covariates, and concurrent depressive symptoms. | Mental Health | CPC | Queensland Department of Families, Youth and Community Care (DFYCC) | 1 (Deterministic) | No |
| **Abajobir et al. (2016)** | Australia | This study examines whether distinct types of childhood maltreatment differentially predict different forms of intimate partner violence | Domestic violence | CPC | Queensland Department of Families, Youth and Community Care (DFYCC) | 1 (Deterministic) | No |
| **Abajobir et al. (2016)** | Australia | This study investigates the association between exposure to prospectively-substantiated childhood maltreatment between 0 to 14 years of age and lifetime cannabis use, abuse and dependence reported at 21 years | Drugs & alcohol | CPC | Queensland Department of Families, Youth and Community Care (DFYCC) | 1 (Deterministic) | No |
| **Abajobir et al. (2017)** | Australia | Determine the association between substantiated childhood maltreatment and injecting drug use | Drugs & Alcohol | CPC | Queensland Department of Families, Youth and Community Care (DFYCC) | 1 (Deterministic) | No |
| **Strathean et al. (2009)** | Australia | Explored whether breastfeeding may protect against maternally-perpetrated child maltreatment. | Child protection | CPC | Queensland Department of Families, Youth and Community Care (DFYCC) | 1 (Deterministic) | No |

(*Continued*)

**Table 2.** (*Continued*)

| Author (Year) | Country | Aims/ Objectives | Research Area | Child Protection Contact (CPC) vs. OHC | Administrative Data Source | Number of administrative datasets (Deterministic/ Probabilistic Linkage) | Linkage Quality (Yes/ No) |
|---|---|---|---|---|---|---|---|
| **Mills et al. (2013)** | Australia | To examine whether notified child maltreatment is associated with adverse psychological outcomes in adolescence, and whether differing patterns of psychological outcome are seen depending on the type of maltreatment. | Mental Health | CPC | Queensland Department of Families, Youth and Community Care (DFYCC) | 1 (Deterministic) | No |
| **Mills et al. (2016)** | Australia | Investigate the incidence of CSA in the same birth cohort using both retrospective self-report and prospective government agency notification, and examine the psychological outcomes in young adulthood. | Mental Health | CPC | Queensland Department of Families, Youth and Community Care (DFYCC) | 1 (Deterministic) | No |
| **Mills et al. (2014)** | Australia | This study examines whether child maltreatment experience predicts adolescent tobacco and alcohol use. The secondary question was whether specific patterns of types of maltreatment were associated with alcohol and/or tobacco use. | Drugs & alcohol | CPC | Queensland Department of Families, Youth and Community Care (DFYCC) | 1 (Deterministic) | No |
| **Mills et al. (2019)** | Australia | to investigate whether child maltreatment is associated with adverse outcomes in cognitive function, high school completion and employment by the age of 21 | Education | CPC | Queensland Department of Families, Youth and Community Care (DFYCC) | 1 (Deterministic) | No |
| **Mills et al. (2017)** | Australia | To investigate whether: (1) child maltreatment is associated with life-time cannabis use, early-onset cannabis use, daily cannabis use and DSM-IV cannabis abuse in young adulthood; and (2) behaviour problems, tobacco use and alcohol use at age 14 are associated with cannabis use. | Drugs & Alcohol | CPC | Queensland Department of Families, Youth and Community Care (DFYCC) | 1 (Deterministic) | No |
| **Parrish et al. (2011)** | Australia | To assess the utility of combining PRAMS data with child protective services (CPS) records to identify risk factors associated with Protective Services Reports (PSR) suggestive of child maltreatment | Child protection | CPC | Alaska's Child Protective Services Agency Register | 1 (Probabilistic) | Yes |
| **Raghavan et al. (2012)** | USA | To estimate the amount of Medicaid expenditures incurred from the purchase of psychotropic drugs–the primary drivers of mental health expenditures among children in the child welfare system | Health insurance | CPC | 1.Medicaid Analytic eXtract (MAX) Research Data Assistance Centre; 2.Child Welfare Agency | 1 (Deterministic & Probabilistic) | Yes |

| Author (Year) | Name of Longitudinal Study | Study Period | Sampling Method | Study Population | | | | Waves in the study: (Age: sample size) | Wave reported: (Age: Sample Size) |
|---|---|---|---|---|---|---|---|---|---|
| | | | | Age at Baseline | Year of birth | Gender-Males (%) | Cohort size at Baseline | | |
| **Egulend et al. (2009)** | Danish longitudinal survey of children (DALSC) | 1995–2007 | NR | Birth | 1995 | NR | 1. Non-CPC (6,000); 2. OHC (1,072); 3. In-home care (1,457) | Wave 1, Baseline: (4 months, n = 6,622); Wave 2: (3.5 years, n = 6,622); Wave 3: (7 years, n = 7,198); Wave 4: (11 years, n = 8,225); Wave 5: (15 years, n = 7,132) | Wave 4: (11 years, Non-welfare children n = 5,242; OHC: n = 433; In-home care: n = 95) |

(*Continued*)

**Table 2.** (*Continued*)

| Author (Year) | Country | Aims/ Objectives | Research Area | Child Protection Contact (CPC) vs. OHC | Administrative Data Source | | Number of administrative datasets (Deterministic/ Probabilistic Linkage) | | Linkage Quality (Yes/ No) |
|---|---|---|---|---|---|---|---|---|---|
| **Hansson et al. (2018)** | Swedish longitudinal Evaluation Through Follow-up (ETF) project | 1971–2001 | Stratified systematic sampling | 9 years | 1972; 1977; 1982; 1987; 1992 | NR | (4,500–12,000)* 5 Cohorts | 1948 Cohort: (12 years, n = 12,000); 1953 Cohort: (12 years, n = 9,000); 1967 Cohort: (12 years, n = 9,000); 1972 Cohort: (9 & 12 years, n = 9,000); 1977 Cohort: (9 & 12 years, n = 4,500); 1982 Cohort: (12 years, n = 9,000); 1987 Cohort: (15 years, n = 9,000); 1992 Cohort: (9 years, n = 9,000) | Wave 1, Baseline (7 years; n = N/A); Wave 2: (9 years; Pooled Data from 5 Cohorts (non-OHC: n = 40,107; OHC: n = 1,482) |
| **Kisely et al. (2019)** | The Mater-University Study of Pregnancy (MUSP) | 1981–2004 | NR | Birth | 1981–1983 | 47% | 7,223 Mother & Child pairs | Wave 1, Baseline: (Mother and child dyads at birth, n = 7,223); Wave 2: (6 months: n = 6,720); Wave 3: (5 years: n = 5,308); Wave 4: (14 years: n = 5,216); Wave 5: (21 years: n = 3,805); Wave 6: (30 years: n = 2,904) | Wave 1, Baseline: (Mother and child dyads at birth, n = 7,223); Wave 4 (14 years: n = NR); Wave 5 (21 years: n = 3,758 & subset n = 2,548) |
| **Kisely et al. (2018)** | The Mater-University Study of Pregnancy (MUSP) | 1981–2004 | NR | Birth | 1981–1983 | 53% | 7,223 Mother & Child pairs | Wave 1, Baseline: (Mother and child dyads at birth, n = 7,223); Wave 2: (6 months: n = 6,720); Wave 3: (5 years: n = 5,308); Wave 4: (14 years: n = 5,216); Wave 5: (21 years: n = 3,805); Wave 6: (30 years: n = 2,904) | Wave 1, Baseline: (Mother and child dyads at birth: n = 7,223); Wave 5 (21 years: n = 3,778) |
| **Kisely et al. (2019)** | The Mater-University Study of Pregnancy (MUSP) | 1981–2004 | NR | Birth | 1981–1983 | 47% | 7,223 Mother & Child pairs | Wave 1, Baseline: (Mother and child dyads at birth: n = 7,223); Wave 2: (6 months: n = 6,720); Wave 3: (5 years: n = 5,308); Wave 4: (14 years: n = 5,216); Wave 5: (21 years: n = 3,805); Wave 6: (30 years: n = 2,904) | Wave 1, Baseline: (Mother and child dyads at birth: n = 7,223); Wave 5 (21 years: n = 3,762) |
| **Olsen et al. (2018)** | Danish longitudinal survey of children (DALSC) | 1995–2011 | NR | Birth | 1995 | 53% | 907 OHC; 5,900 non-OHC | Wave 1, Baseline: (4 months, n = 6,622); Wave 2: (3.5 years: n = 6,622); Wave 3: (7 years: n = 7,198); Wave 4: (11 years: n = 8,225); Wave 5: (15 years: n = 7,132); Wave 6: (18 years: n = 5,139) | Wave 1, Baseline: (Birth, OHC: n = 907, non-OHC: n = 5,900); Wave 5: (15 years: OHC: n = 169, non-OHC: n = 4,568); Wave 6: (18 years: OHC: n = 817, non-OHC: n = 4,322) |

(*Continued*)

**Table 2.** (Continued)

| Author (Year) | Country | Aims/ Objectives | Research Area | Child Protection Contact (CPC) vs. OHC | Administrative Data Source | | Number of administrative datasets (Deterministic/ Probabilistic Linkage) | | Linkage Quality (Yes/ No) |
|---|---|---|---|---|---|---|---|---|---|
| **Parrish et al. (2016)** | Alaska Pregnancy Risk Assessment Monitoring System (PRAMS) | 2009–2014 | Stratified systematic sampling | Birth | 2009–2010 | NR | 2,389 | 1990–2016 Cohorts: (Annual sample sizes per state range from about 1000 to 3000 women) | Wave 1: (Birth-2 years: n = 2,389) |
| **Parrish et al. (2017)** | Alaska Pregnancy Risk Assessment Monitoring System (PRAMS) | 2009–2014 | Stratified systematic sampling | Birth | 2009–2011 | NR | 1,235 | 1990–2016 Cohorts: (Annual sample sizes per state range from about 1000 to 3000 women) | Wave 1: (Birth: n = 1,235) |
| **Raghavan et al. (2017)** | National Survey of Child and Adolescent Well-Being (NSCAW) | 1999–2003 | NR | NR | NR | NR | Child Protection Contact (CPC) (5,501); Long term foster care placement (LTFC) (727) | Wave 1: (Birth: n = 6,228); Wave 2: (9 years: n = 5,873); Wave 3: (14 years: n = NR) | Pooled (Wave 1-wave 3) sample: (CPS: n = 2,309, LTFC: n = 423) |
| **Sidebotham et al. (2000)** | The Avon Longitudinal Study of Parents and Children (ALSPAC) | 1991–1998 | NR | Pre-birth | 1991–1992 | NR | 14,451 | Wave 1: (Pre-birth, n = 14,893); Wave 2: (1 month: n = 14,256); Wave 3: (6–8 months: n = 11,194, Partner = 6,861); Wave 4: (18 months: n = 10,750); Wave 5: (21 months: n = 10,323); Wave 6: (30 months: n = 10,289); Wave 7: (33 months: n = 9,635) | Wave 3: (8 months, n = 11,194, Partner: n = 6,861); Wave 4: (18 months, n = 10,750); Wave 5: (21 months, n = 10,323); Wave 6: (30 months, n = 10,289); Wave 7: (33 months, n = 9,635) |
| **Sidebotham et al. (2003)** | The Avon Longitudinal Study of Parents and Children (ALSPAC) | 1991–1998 | NR | 1 month | 1991–1992 | (56% registered & 52% non-registered) | 14,256 | Wave 1: (Pre-birth, n = 14,893); Wave 2: (1 month: n = 14,256); Wave 3: (6–8 months: n = 11,194, Partner = 6,861); Wave 4: (18 months: n = 10,750); Wave 5: (21 months: n = 10,323); Wave 6: (30 months: n = 10,289); Wave 7: (33 months: n = 9,635) | Wave 2: (1 month, n = 14,256); Wave 6: (30 months, n = 115 registered vs n = 14,105 non-registered children) |
| **Sidebotham et al. (2006)** | The Avon Longitudinal Study of Parents and Children (ALSPAC) | 1991–1998 | NR | Pre-birth | 1991–1992 | NR | 14,256 | Wave 1: (Pre-birth, n = 14,893); Wave 2: (1 month: n = 14,256); Wave 3: (6–8 months: n = 11,194, Partner = 6,861); Wave 4: (18 months: n = 10,750); Wave 5: (21 months: n = 10,323); Wave 6: (30 months: n = 10,289); Wave 7: (33 months: n = 9,635) | Wave 2: (One month: n = 14,256); Wave 7: (36 months: n = NR) |

(*Continued*)

**Table 2.** (Continued)

| Author (Year) | Country | Aims/ Objectives | Research Area | Child Protection Contact (CPC) vs. OHC | Administrative Data Source | | Number of administrative datasets (Deterministic/ Probabilistic Linkage) | | Linkage Quality (Yes/ No) |
|---|---|---|---|---|---|---|---|---|---|
| **Sidebotham et al. (2002)** | The Avon Longitudinal Study of Parents and Children (ALSPAC) | 1991–1998 | NR | Pre-birth | 1991–1992 | 52% | 14,256 | Wave 1: (Pre-birth: n = 14,893); Wave 2: (1 month: n = 14,256); Wave 3: (6–8 months: n = 11,194, Partner = 6,861); Wave 4: (18 months: n = 10,750); Wave 5: (21 months: n = 10,323); Wave 6: (30 months: n = 10,289); Wave 7: (33 months: n = 9,635) | Wave 2: (One month: n = 14,256); Wave 3: (8 months: n = 11,194); Wave 5: (21 months: n = 10,323); Wave 7: (33 months: n = 9,635) |
| **Teyhan et al. (2019)** | The Avon Longitudinal Study of Parents and Children (ALSPAC) | 1991–2009 | NR | Pre-birth | 1991–1992 | (50% (No CLA/CIN); 48% CIN; 51% CLA) | 14,868 | Wave 1: (Pre-birth: n = 14,893); Wave 2: (1 month: n = 14,256); Wave 3: (6–8 months: n = 11,194, Partner = 6,861); Wave 4: (18 months: n = 10,750); Wave 5: (21 months: n = 10,323); Wave 6: (30 months: n = 10,289); Wave 7: (33 months: n = 9,635) | Wave 3: (1 year: n = 13,988); Wave 8: (7–18 years, Booster: n = 718); Wave 9: (>18 years, Booster: n = 183) |
| **Austin et al. (2019)** | Alaska Longitudinal Child Abuse and Neglect Linkage (ALCANLink) project & PRAMS | 2009–2014 | Stratified systematic sampling | Birth | 2009–2011 | (53% AN & 49% NN) | AN (1,257); NN (2,102) | 1990–2016 Cohorts: (Birth, n = 1,000–3,000) | Wave 1: (Birth -5/6 years) |
| **Austin et al. (2018)** | Alaska Longitudinal Child Abuse and Neglect Linkage (ALCANLink) project & PRAMS | 2009–2015 | Stratified systematic sampling | Birth | 2009–2011 | 51% | 3,549 | 1990–2016 Cohorts: (Birth, n = 1,000–3,000) | Wave 1 (Birth -5/6 years) |
| **Hansson et al. (2020)** | Swedish longitudinal Evaluation Through Follow-up (ETF) project | NR | Stratified systematic sampling | 9 years | 1972; 1977; 1982; 1987; 1992 | NR | (4,500–12,000)* 5 Cohorts | 1948 Cohort: (12 years, n = 12,000); 1953 Cohort: (12 years, n = 9,000); 1967 Cohort: (12 years, n = 9,000); 1972 Cohort: (9 & 12 years, n = 9,000); 1977 Cohort: (9 & 12 years, n = 4,500); 1982 Cohort: (12 years, n = 9,000); 1987 Cohort: (15 years, n = 9,000); 1992 Cohort: (9 years, n = 9,000) | Wave 2: (9 years, n = NR); Wave 3: (12 years, n = NR) |
| **Abajobir et al. (2017)** | The Mater-University Study of Pregnancy (MUSP) | 1981–2004 | NR | Birth | 1981–1983 | 47% | 7,223 Mother & Child pairs | Wave 1, Baseline: (Mother and child dyads at birth: n = 7,223); Wave 2: (6 months: n = 6,720); Wave 3: (5 years: n = 5,308); Wave 4: (14 years: n = 5,216); Wave 5: (21 years: n = 3,805); Wave 6: (30 years: n = 2,904) | Wave 1, Baseline: (Mother and child dyads at birth, n = 7,223); Wave 3: 5 years; Wave 4 (14 years: n = NR); Wave 5 (21 years: n = 3,752) |

(*Continued*)

**Table 2.** (Continued)

| Author (Year) | Country | Aims/ Objectives | Research Area | Child Protection Contact (CPC) vs. OHC | Administrative Data Source | | Number of administrative datasets (Deterministic/ Probabilistic Linkage) | | Linkage Quality (Yes/ No) |
|---|---|---|---|---|---|---|---|---|---|
| **Abajobir et al. (2017)** | The Mater-University Study of Pregnancy (MUSP) | 1981–2004 | NR | Birth | 1981–1983 | 50% | 7,223 Mother & Child pairs | Wave 1, Baseline: (Mother and child dyads at birth: n = 7,223); Wave 2: (6 months: n = 6,720); Wave 3: (5 years: n = 5,308); Wave 4: (14 years: n = 5,216); Wave 5: (21 years: n = 3,805); Wave 6: (30 years: n = 2,904) | Wave 1, Baseline: (Mother and child dyads at birth, n = 7,223); Wave 3: (5 years: n = NR); Wave 4 (14 years: n = NR); Wave 5 (21 years: n = 3,730) |
| **Abajobir et al. (2016)** | The Mater-University Study of Pregnancy (MUSP) | 1981–2004 | NR | Birth | 1981–1983 | 45% | 7,223 Mother & Child pairs | Wave 1, Baseline: (Mother and child dyads at birth: n = 7,223); Wave 2: (6 months: n = 6,720); Wave 3: (5 years: n = 5,308); Wave 4: (14 years: n = 5,216); Wave 5: (21 years: n = 3,805); Wave 6: (30 years: n = 2,904) | Wave 1, Baseline: (Mother and child dyads at birth: n = 7,223); Wave 4 (14 years: n = NR); Wave 5 (21 years: n = 3,322) |
| **Abajobir et al. (2016)** | The Mater-University Study of Pregnancy (MUSP) | 1981–2004 | NR | Birth | 1981–1983 | 48% | 7,223 Mother & Child pairs | Wave 1, Baseline: (Mother and child dyads at birth: n = 7,223); Wave 2: (6 months: n = 6,720); Wave 3: (5 years: n = 5,308); Wave 4: (14 years: n = 5,216); Wave 5: (21 years: n = 3,805); Wave 6: (30 years: n = 2,904) | Wave 1, Baseline: (Mother and child dyads at birth: n = 7,223); Wave 4 (14 years: n = NR); Wave 5 (21 years: n = 2,526) |
| **Abajobir et al. (2017)** | The Mater-University Study of Pregnancy (MUSP) | 1981–2004 | NR | Birth | 1981–1983 | 47% | 7,223 Mother & Child pairs | Wave 1, Baseline: (Mother and child dyads at birth: n = 7,223); Wave 2: (6 months: n = 6,720); Wave 3: (5 years: n = 5,308); Wave 4: (14 years: n = 5,216); Wave 5: (21 years: n = 3,805); Wave 6: (30 years: n = 2,904) | Wave 1, Baseline: (Mother and child dyads at birth, n = 7,223); Wave 5: (21 years: n = 3,750) |
| **Strathean et al. (2009)** | The Mater-University Study of Pregnancy (MUSP) | 1981–2000 | NR | Birth | 1981–1983 | 52% | 7,223 Mother & Child pairs | Wave 1, Baseline: (Mother and child dyads at birth: n = 7,223); Wave 2: (6 months: n = 6,720); Wave 3: (5 years: n = 5,308); Wave 4: (14 years: n = 5,216); Wave 5: (21 years: n = 3,805); Wave 6: (30 years: n = 2,904) | Wave 1, Baseline: (Mother and child dyads at birth, n = 7,223); Wave 2: (6 months: n = 6,621); Wave 4: (15 years: n = 5,890) |

(*Continued*)

**Table 2.** (*Continued*)

| Author (Year) | Country | Aims/ Objectives | Research Area | Child Protection Contact (CPC) vs. OHC | Administrative Data Source | | Number of administrative datasets (Deterministic/ Probabilistic Linkage) | | Linkage Quality (Yes/ No) |
|---|---|---|---|---|---|---|---|---|---|
| **Mills et al. (2013)** | The Mater-University Study of Pregnancy (MUSP) | 1981–2000 | NR | Birth | 1981–1983 | 52% | 7,223 Mother & Child pairs | Wave 1, Baseline: (Mother and child dyads at birth: n = 7,223); Wave 2: (6 months: n = 6,720); Wave 3: (5 years: n = 5,308); Wave 4: (14 years: n = 5,216); Wave 5: (21 years: n = 3,805); Wave 6: (30 years: n = 2,904) | Wave 1, Baseline: (Mother and child dyads at birth, n = 7,223; Wave 4: (14 years: n = 5,172) |
| **Mills et al. (2016)** | The Mater-University Study of Pregnancy (MUSP) | 1981–2004 | NR | Birth | 1981–1983 | 52% | 7,223 Mother & Child pairs | Wave 1, Baseline: (Mother and child dyads at birth: n = 7,223); Wave 2: (6 months: n = 6,720); Wave 3: (5 years: n = 5,308); Wave 4: (14 years: n = 5,216); Wave 5: (21 years: n = 3,805); Wave 6: (30 years: n = 2,904) | Wave 1, Baseline: (Mother and child dyads at birth, n = 7,223; Wave 5: (21 years: n = 3,739) |
| **Mills et al. (2014)** | The Mater-University Study of Pregnancy (MUSP) | 1981–2000 | NR | Birth | 1981–1983 | 52% | 7,223 Mother & Child pairs | Wave 1, Baseline: (Mother and child dyads at birth: n = 7,223); Wave 2: (6 months: n = 6,720); Wave 3: (5 years: n = 5,308); Wave 4: (14 years: n = 5,216); Wave 5: (21 years: n = 3,805); Wave 6: (30 years: n = 2,904) | Wave 1, Baseline: (Mother and child dyads at birth: n = 7,223; Wave 4: (14 years: n = 5,200) |
| **Mills et al. (2019)** | The Mater-University Study of Pregnancy (MUSP) | 1981–2004 | NR | Birth | 1981–1983 | NR | 7,223 Mother & Child pairs | Wave 1, Baseline: (Mother and child dyads at birth: n = 7,223); Wave 2: (6 months: n = 6,720); Wave 3: (5 years: n = 5,308); Wave 4: (14 years: n = 5,216); Wave 5: (21 years: n = 3,805); Wave 6: (30 years: n = 2,904) | Wave 1, Baseline: (Mother and child dyads at birth: n = 7,223; Wave 5: (21 years: n = 3,778) |
| **Mills et al. (2017)** | The Mater-University Study of Pregnancy (MUSP) | 1981–2004 | NR | Birth | 1981–1983 | 47% | 7,223 Mother & Child pairs | Wave 1, Baseline: (Mother and child dyads at birth: n = 7,223); Wave 2: (6 months: n = 6,720); Wave 3: (5 years: n = 5,308); Wave 4: (14 years: n = 5,216); Wave 5: (21 years: n = 3,805); Wave 6: (30 years: n = 2,904) | Wave 1, Baseline (Mother and child dyads at birth: n = 7,223); Wave 4: (14 years: n = NR); Wave 5: (21 years: n = 3,778) |

(*Continued*)

**Table 2.** (Continued)

| Author (Year) | Country | Aims/Objectives | Research Area | Child Protection Contact (CPC) vs. OHC | Administrative Data Source | | | Number of administrative datasets (Deterministic/Probabilistic Linkage) | Linkage Quality (Yes/No) |
|---|---|---|---|---|---|---|---|---|---|
| **Parrish et al. (2011)** | Alaska Pregnancy Risk Assessment Monitoring System (PRAMS) | 1997–2004 | Stratified systematic sampling | Birth | 1997–1999 | 48% | 5, 421 | 1990–2016 Cohorts: (Annual sample sizes per state range from about 1000 to 3000 women) | Wave 1, Baseline (Birth: n = 5,421); Wave 2: (48 months: n = 4,217) |
| **Raghavan et al. (2012)** | National Survey of Child and Adolescent Well-Being (NSCAW) | 1999–2003 | NR | 2 years | NR | 48% | NSCAW (2,831); Matched child observations (2,821) | Wave 1: (Birth: n = 6,228); Wave 2: (9 years: n = 5,873); Wave 3: (14 years: n = NR) | Pooled (Wave 1-wave 4): n = 5,652 |

| Author (Year) | Timeframe between reported waves (months) | Outcome Measures | | Missing data (Yes/No) | Attrition rate | Described attrition (Yes/No) | Corrected attrition (Yes/No) | Attrition analysis (Yes/No) | Selection bias (Yes/No) | Sensitivity analysis Yes/No |
|---|---|---|---|---|---|---|---|---|---|---|
| | | Standardized | Non-standardized | | | | | | | |
| **Egulend et al. (2009)** | 36 months | 1. Strengths and Difficulties screening (SDQ) for mental health 2. ICD-10 Psychiatric diagnosis | 1. School performance and satisfaction; 2. Leisure activities | Yes | NR | Yes | Yes | No | No | No |
| **Hansson et al. (2018)** | Waves 1–2 = 24 months | Cognitive Test Scores | Academic achievement | Yes | NR | No | Yes | No | Yes | No |
| **Kisely et al. (2019)** | (Waves 1–4 = 168 months); Waves 4–5 = 84 months) | 1. WHO (CIDI-DSM-IV) scale for Nicotine use, dependence & withdrawal; 2. Depression (CES-D) scale | 1. Prevalence of smoking; 2. Persistent smoking | Yes | 48% | Yes | Yes | Yes | No | Yes |
| **Kisely et al. (2018)** | (Waves 1–5 = 252 months) | 1. Centre for Epidemiological Studies-Depression scales (CES-D) 2. Achenbach Youth Self-Report (YASR) scale; 3. WHO (CIDI-DSM-IV) scale | None | Yes | 48% | Yes | Yes | Yes | No | Yes |
| **Kisely et al. (2019)** | (Waves 1–5 = 252 months) | WHO (CIDI-DSM-IV) scale for alcohol use and dependence | Alcohol use in the last month | Yes | 48% | Yes | Yes | Yes | No | No |
| **Olsen et al. (2018)** | (Waves 1–2 = 180 months); Waves 2–3 = 36 months) | None | 1. Self-perceived academic ability (SAA) 2. Staying-on rates | Yes | NR | Yes | No | Yes | No | No |
| **Parrish et al. (2016)** | N/A | None | Maltreatment report to Child Protective Services | Yes | N/A | No | No | No | No | Yes |
| **Parrish et al. (2017)** | N/A | None | Child maltreatment | Yes | NR | Yes | Yes | Yes | Yes | No |
| **Raghavan et al. (2017)** | Wave 1- Wave 3 = 36 months | None | Ascertainment of foster care status | Yes | NR | No | No | No | Yes | No |
| **Sidebotham et al. (2000)** | (Waves 3–4 = 10 months); (Waves 4–5 = 3 months); (Waves 5–6 = 9 months); (Waves 6–7 = 3 months) | None | Child abuse investigations and registrations | No | NR | No | No | No | No | No |
| **Sidebotham et al. (2003)** | (Waves 2–6 = 29 months) | None | Child protection registration | Yes | NR | Yes | No | No | Yes | No |
| **Sidebotham et al. (2006)** | Wave 2–7: 35 months | None | 1. Investigation for suspected maltreatment; 2. Registration on the child protection register | Yes | NR | Yes | No | No | Yes | No |
| **Sidebotham et al. (2002)** | (Waves 2–3 = 7 months); (Waves 3–5 = 13 months); (Waves 5–7 = 12 months) | None | Child abuse registration | Yes | NR | No | No | No | Yes | No |

(*Continued*)

**Table 2.** (*Continued*)

| Author (Year) | Country | Aims/ Objectives | Research Area | Child Protection Contact (CPC) vs. OHC | Administrative Data Source | | Number of administrative datasets (Deterministic/ Probabilistic Linkage) | | Linkage Quality (Yes/ No) | |
|---|---|---|---|---|---|---|---|---|---|---|
| **Teyhan et al. (2019)** | (Waves 3–8 = 84 months); (Waves 8–9 = 132 months) | None | 1. Educational attainment; 2. Persistent absence from school; 3. Special educational needs (SEN) status; 4. School Mobility | Yes | NR | No | No | No | No | Yes |
| **Austin et al. (2019)** | Wave 1 (5/6 years) | None | Child Protective Service Contact | Yes | NR | No | No | No | No | No |
| **Austin et al. (2018)** | Wave 1 (5/6 years) | None | Age at first CP contact | Yes | NR | No | No | No | No | Yes |
| **Hansson et al. (2020)** | Waves 2–3 = 36 months | None | Cognitive ability | Yes | NR | No | No | No | No | No |
| **Abajobir et al. (2017)** | (Waves 1–2 = 6 months); (Waves 2–3 = 54 months); (Waves 3–4 = 108 months); (Waves 4–5 = 84 months) | 1. Achenbach's YASR Behaviour Checklist (Auditory & Visual Hallucinations); 2. Peter's Delusional Inventory (PDI); 3. WHO (CIDI-DSM-IV) scale for diagnoses of psychosis | None | Yes | 48% | Yes | Yes | Yes | Yes | Yes |
| **Abajobir et al. (2017)** | (Waves 1–2 = 6 months); (Waves 2–3 = 54 months); (Waves 3–4 = 108 months); (Waves 4–5 = 84 months) | 1. Achenbach's Young Adult Self-Report (YASR) Behaviour Checklist (4 items); 2. Centre for Epidemiological Studies Depression Scale (CES-D) | QoL Self Report (Happy/ Satisfaction scales) | Yes | 48% | Yes | Yes | Yes | No | No |
| **Abajobir et al. (2016)** | (Waves 1–2 = 6 months); (Waves 2–3 = 54 months); (Waves 3–4 = 108 months); (Waves 4–5 = 84 months) | 1. Composed abuse scale (CAS) 2. Child Behaviour Checklist (CBCL) 3. Life events scale; 4. Conflict tactics scale | None | Yes | 54% | Yes | Yes | Yes | No | Yes |
| **Abajobir et al. (2016)** | (Waves 1–2 = 6 months); (Waves 2–3 = 54 months); (Waves 3–4 = 108 months); (Waves 4–5 = 84 months) | WHO (CIDI-DSM-IV) scale for Lifetime cannabis abuse and dependence | Early age of onset of cannabis abuse | Yes | 65% | Yes | Yes | Yes | Yes | No |
| **Abajobir et al. (2017)** | (Waves 1–5 = 252 months) | Depression: Delusions-Symptoms-States Inventory scale (DSSI) | Ever injected illicit drugs | Yes | 48% | Yes | Yes | Yes | Yes | Yes |
| **Strathean et al. (2009)** | (Waves 1–3 = 6 months); Waves 3–4 = 174 months | Depression: Delusions-Symptoms-States Inventory scale (DSSI) | Child maltreatment | Yes | 18% | Yes | Yes | Yes | Yes | Yes |
| **Mills et al. (2013)** | (Waves 1–4 = 168 months) | Achenbach Youth Self-Report (YSR) questionnaires | None | Yes | 28% | Yes | No | Yes | No | Yes |
| **Mills et al. (2016)** | (Waves 1–5 = 252 months) | WHO (CIDI-DSM-IV) scale for psychological outcomes at age 21 | None | Yes | 48% | Yes | Yes | Yes | No | Yes |
| **Mills et al. (2014)** | (Waves 1–4 = 168 months) | None | 1. Smoking status; 2. Alcohol use | Yes | 28% | Yes | No | Yes | No | Yes |
| **Mills et al. (2019)** | (Waves 1–5 = 252 months) | Peabody Picture Vocabulary Test (PPVT) | 1. Failure to complete high school; 2. Failure to be employed or education at 21 years | No | 48% | Yes | No | No | No | No |

(*Continued*)

**Table 2.** (Continued)

| Author (Year) | Country | Aims/ Objectives | Research Area | Child Protection Contact (CPC) vs. OHC | Administrative Data Source | | Number of administrative datasets (Deterministic/ Probabilistic Linkage) | | Linkage Quality (Yes/ No) | |
|---|---|---|---|---|---|---|---|---|---|---|
| **Mills et al. (2017)** | (Waves 1–5 = 252 months) | 1. WHO (CIDI-DSM-IV) scale for Cannabis use/ dependence; 2. Achenbach Child Behaviour Checklist (CBCL); 3. Delusions–Symptoms–States Inventory (DSSI) | Self-report | Yes | 48% | Yes | No | No | No | No |
| **Parrish et al. (2011)** | (Waves 1–2 = 48 months) | None | Protective service report | No | 22% | No | No | No | No | No |
| **Raghavan et al. (2012)** | Wave 1- Wave 4 = 48 months | Internalizing or externalizing scales of the CBCL | 1. Non-zero Medicaid expenditures in a calendar year; 2. Mean total annual Medicaid expenditure per child | No | NR | No | No | No | No | Yes |

Notes

**CIDI** Composite International Diagnostic Interview

**CPC** Child Protection Contact

**CPS** Child Protective Services

**CSA** Child Sexual Abuse

**DSM-IV** Diagnostic and Statistical Manual of Mental Disorders, 4th edition

**DVSA** Domestic violence and sexual assault

**IPV** Intimate Partner Violence

**LTFC** Long Term Foster Care

**N/A** Not Applicable

**NR** Not Reported

**OHC** Out-of-home care

**SDQ** Strength and Difficulties Questionnaire

**WHO** World Health Organisation

**YASR** Young Adult Self Report

The cohort sizes ranged from 1,200 children to approximately 14, 000 children. Most studies (83%) reported only one administrative database that was integrated with the longitudinal data, while 17% had multiple datasets linked and these ranged from census data, psychiatric registers, educational databases, medical aid data, child birth and death reviews. Almost all (97%) of the studies reported a state-wide child protection dataset integrated with the longitudinal data. About 23% of studies from two longitudinal studies reported systematic random sampling method. These studies were the Alaska Pregnancy Risk Assessment Monitoring System (PRAMS) and the Evaluation through Follow-up (ETF) studies.

GUILD [7] recommend reporting on the following three aspects when reporting on studies using linked datasets: i) description of the population included in the data set i.e. how the data were generated, processed and quality controlled, ii) data linkage processes, and; iii) quality of data linkage including accounting for linkage error. Most studies only reported on one of the steps which is the data linkage method used. Fifty seven percent reported using a deterministic linkage method which mainly involved using a unique personal identification number to link datasets. This linkage method is well established in Scandinavian countries [24, 83], and is increasingly becoming common in other countries. Only two studies reported using probabilistic matching, which involves using a set of non-unique identifiers to link data [84]. Two

studies [55, 85] reported using a combination of probabilistic and deterministic methods and nine studies did not report on any linkage methods.

Only four studies reported on the linkage quality. Parrish, Young [86] reported on the proportion of successful matches, manual review of suspected matches that met a certain probability score threshold, [55] while two studies from Raghavan, Brown [85] and reported on the number of records that were linked and unlinked from the source file including statistical differences in linked and unlinked data on key variables.

## Biases reported

There are several biases which commonly occur in longitudinal studies [47]. However, for the purposes of this review we report on three of the most common occurring biases, attrition, missing data and selection bias.

**Missing data.** Incomplete data is common in longitudinal research, as reflected in this review where missing data were reported in 87% of the studies (Table 3). In the past, three traditional mechanisms of missing data were reported [87]. When missingness is unrelated to the data, this is termed missing completely at random (MCAR), while if the probability of missing data on a variable is unrelated to the value of that variable itself but may be related to the values of other variables in the dataset this is referred to as missing at random (MAR). A mechanism which should not be ignored in longitudinal analysis is termed missing not at random (MNAR) [87, 88]. This refers to missingness that is contingent on the unobserved data, as reported in studies where there was an over-representation of children exposed to child protection agencies with missing data resulting in over-estimation of outcomes in this group compared to the general population [89, 90] and also missing data due to attrition.

Studies in this review reported missing data on certain covariates (MCAR) such as child maltreatment, parental race, paternal income and education and breastfeeding status [47, 52, 81, 91–96]. Missing data were also reported on outcome variables such as those from the Strengths and Difficulties Questionnaire [24]. There are a range of simple to more sophisticated analytical methods of handling missing data that can be applied to reduce bias in reported outcomes. The simplest method reported was listwise deletion [4, 21, 59, 97, 98] and including missing data as a separate category for each covariate in regression analysis (Missing Indicator Method) [47, 81, 93–96]. Sophisticated methods included multiple imputation using Markov chain iterative regression methods (MCMC) [94], multiple imputation using chained equations (MICE) [45], and multiple imputation using the fully conditional specification (FCS) method [99] (S3 Table).

**Missing data due to attrition.** Attrition) is a type of missingness that can occur in longitudinal studies, which typically occurs due to loss to follow up, death, emigration or nonreturn of a survey and withdrawal from the study [100]. Attrition rates were reported for 53%

**Table 3. Biases reported.**

| Type of Bias | N (Number of studies) | % |
|---|---|---|
| Missing data | 26 | 87% |
| Attrition rate | - | 18–65% |
| Described attrition | 19 | 63% |
| Corrected attrition | 12 | 40% |
| Analysis of attrition | 14 | 47% |
| Selection bias | 10 | 33% |
| Sensitivity Analysis | 13 | 43% |

of the studies and the rates ranged from 18% to 65% (Table 3). Even though the attrition rate was not mentioned in almost half of the studies, attrition was described for 63% of all studies. The review identified attrition as occurring due to loss of follow-up or differential attrition occurring among families with reported cases of substantiated maltreatment, those from higher socio-economic disadvantaged backgrounds and among males and indigenous people (particularly among MUSP studies) [4, 21, 46, 82, 97, 98, 101, 102]. Other attrition reported was death or early infant loss [47, 55, 93, 96], non-response [47] and emigration [47, 55].

Forty seven percent (47%) of all studies mentioned that they conducted some attrition analysis, while 40% reported some methods of correcting attrition loss. While these methods were described in the studies, the analysis output was not shown for all studies. Attrition analysis was conducted to determine if there would be any significant differences in outcomes among participants lost to follow up and those remaining in the study. The main methods of correcting for attrition were inverse probability weighting [46, 58, 59, 81, 101, 103, 104] and propensity score analysis [21, 97, 98], while no specific method was described in some studies [24]. Inverse probability weighting was conducted to the analysis of subjects remaining in the cohort to adjust for loss to follow up to the included subjects to restore the representation of subjects. Propensity score analysis was conducted to determine the impact of differential attrition by inclusion of a weighted variable which takes account of baseline covariates.

**Selection bias.** Selection bias occurs when there is a systematic difference between those who participate in the study and those who do not (affecting generalisability) [105, 106]. Selection bias was reported for 33% of the studies (Table 3). Selection bias may result in over-estimation of outcomes among young people exposed to child protection compared with young people in the general population [89]. Restricting the study to certain population groups which may not be representative of the entire population of interest may lead to selection bias [55, 85]. In addition, selection bias also occurs if a population of interest possesses certain unique characteristics giving them a higher chance of recruitment to a study compared to the population without those characteristics [93, 95, 96]. Some authors reported conducting weighted analysis in order to account for potential selection bias [46, 103, 104].

**Sensitivity analysis.** Sensitivity analysis is conducted to determine if small changes in exposure or confounding variables alter the significance of reported outcomes in situations where there could be potential measurement errors [107]. Sensitivity analysis was reported for 43% of the studies, but only eight out of the thirteen studies reported the actual method of analysis conducted. Sensitivity analysis was conducted through modifying some covariates, such as child maltreatment, by expanding the definition to include or exclude notified or suspect cases of maltreatment and through measuring multiple forms versus a single form of abuse [21, 52, 58, 59, 81, 104].

Other authors also reported restricting the analysis to groups of people with certain characteristics [45] or adding [94] or removing [81] one or more covariates to the analysis in order to reduce bias. Addition of covariates at subsequent waves resulted in either strengthening, weakening or no change to the effect sizes in some studies [99]. The main sensitivity analysis methods presented in the eight studies were logistic regression [21, 45, 58, 59, 81, 98, 102] and multiple regression analysis [52] controlling for known confounders and effect modifiers (S3 Table).

## Statistical methods

There were two groups of statistical methods identified in the study. These included data preparation methods and the main statistical analysis method reported.

**Data preparation methods.** Most authors conducted some preliminary data preparation, descriptive or bivariate analysis to address missing data and identify significant covariates to

include as confounders in final in multivariate models. Multiple data preparation methods were described and ranged from descriptive statistics to bivariate and simple regression analysis (S3 Table). In addition, multiple imputation, data weighting and propensity analysis procedures were applied to correct for missing data. Some authors did not provide full details of the analytical methods used to correct for missing data. Common descriptive parameters were frequencies, percentages, means, incidence rates and population attributable risk. Chi-square tests (53%) were also commonly reported as a method to determine association of confounders and outcome variables. Other methods included two-sample *t*-tests (13%), correlation analysis (7%) and to a lesser extent, concordance analysis (3%), logistic regression (3%), and cumulative risk factor analysis (3%).

**Main analytical method.** The main method of analysis for each study was identified. These are shown in Table 4. The main analytical method reported by most studies was logistic regression (63%) followed by multiple regression methods (10%). Logistic regression methods were used for analysing risk factors and associated outcomes, attrition analysis and sensitivity analysis. Advanced analytical methods included generalised linear models (GLM) [108], multinomial logistic regression using Vermunt's three step Latent Class Analysis approach and Growth Mixture Modelling [92], and survival analysis using Kaplan-Meier, Cox (proportional hazards) regression and Nelson-Aalen Estimation methods [55, 99]. A few studies used a combination of methods, where in most cases logistic regression was included as one of the main methods [45, 47, 55, 82]. Only one study reported descriptive statistics as their main method of analysis [109].

The main outcomes evaluated in the studies were standardised and self-reported measures from the main research areas reported in Table 5. There were some notable similarities of reported confounding variables across all studies and most of them (93%) used individual and family characteristics as confounders. These included early childhood experiences, sociodemographic variables, pre-natal exposure and parental (mostly maternal) risk factors. Five studies reported on potential mediating variables, these included school mobility [47, 89], parenting age, education, psychiatric history and poverty [93], gender [46], young people's income, education, marital status, neighbourhood characteristics [21], smoking and alcohol use [97, 102], receipt of social welfare, education and marital status [104], race and receipt of public aid [86]. One study [94] found that parenting and social stress did not moderate the relationship between intimate partner violence and maltreatment. One study reported [98] the following as potential mediating variables: receipt of social welfare, the young person's educational achievement, and the young person's marital status. Only three studies [47, 90, 92] reported some a*ssumptions* of *statistical tests such as tests for normality and homogeneity in variances before conducting data analysis.*

## Quality assessment

The Kmet, GUILD and RECORD checklists were used to rate the methodological quality of included studies. The results of the quality assessment are shown in Table 6. Based on the "QualSyst" Standard Quality assessment for evaluating primary research papers by Kmet, Cook [78], the final quality scores ranged from 55% (adequate quality) to 100% (Strong quality) with a median score of 91%, indicating high quality across all studies reviewed. The final quality scores for the GUILD and RECORD checklist ranged from 10% to 79% and only three studies had scores greater than 50%. The median score was 23%, indicating poor quality across all studies reviewed. The inter-rater reliability test was 81% (95%CI: 75%; 88%) for the Kmet scores and 77% (95%CI: 70%; 85%) for the GUILD and RECORD scores.

**Table 4. Main statistical method.**

| Author | Domain & Analysis Procedure | Statistical parameters | Assumption test | Independent Variables | Mediation and Moderating Variables |
|---|---|---|---|---|---|
| **Egulend et al, (2009)** | Regression Analysis<br>Logistic Regression | Odds ratios, 5% significance level | NR | Individual, family | NR |
| **Hansson et al. (2018)** | Regression Analysis<br>Multiple Regression Analysis | Beta, standard errors, t-statistic, significance level | NR | Individual, family | Mediating:<br>School change |
| **Kisely et al. (2019)** | Regression Analysis<br>Logistic Regression | Odds ratios, 95% CIs, p-values | NR | Individual, family | Mediating:<br>Alcohol use and depression |
| **Kisely et al. (2018)** | Regression Analysis<br>Logistic Regression | Odds ratios, 95% CIs, p-values | NR | Individual, family, community | Mediating:<br>Income, education, Marital status, Characteristics of neighbourhood |
| **Kisely et al. (2019)** | Regression Analysis<br>Logistic Regression | Odds ratios, 95% CIs, p-values | NR | Individual, family, community | NR |
| **Olsen et al. (2018)** | Regression Analysis<br>Multiple Regression Analysis &<br>Linear Probability Model | Unstandardized beta, P-values, adjusted R-squared, standard errors, Significance testing p values (95%, 99%, and 90%) | NR | Individual, family | Mediating:<br>School change |
| **Parrish et al. (2016)** | Regression Analysis<br>Logistic Regression | Frequencies, percentages, odds rations, 95% CI | NR | Individual, family | Moderating:<br>Parenting and social stress |
| **Parrish et al. (2017)** | 1. Regression Analysis<br>Logistic Regression<br>2. Survival Analysis<br>Nelson-Aalen Estimation | 1. Odds ratios, confidence intervals, p-values;<br>2. Weighted Aalen hazard-based estimation, incidence proportion, frequency counts, weighted proportions, Hazard ratios, 95% CI, p-values | NR | Individual, family | NR |
| **Raghavan et al. (2017)** | Regression Analysis<br>Logistic Regression | Odds ratios, standard errors, p-vales | NR | Individual | NR |
| **Sidebotham et al. (2000)** | Descriptive Analysis | Frequencies, Percentages, Incidence rate/ 10,000 children | N/A | Individual, family | N/A |
| **Sidebotham et al. (2003)** | Regression Analysis<br>Logistic Regression | Odds ratios, standard errors, p-vales | NR | Individual, family | NR |
| **Sidebotham et al. (2006)** | Regression Analysis<br>Logistic Regression | Odds ratios, 95% CIs, p-values | NR | Individual, family, community | Mediating:<br>Age at parenting, education, psychiatric history, poverty |
| **Sidebotham et al. (2002)** | Regression Analysis<br>Logistic Regression | Odds ratios, 95% CIs | NR | Individual, family | NR |
| **Teyhan et al. (2019)** | Regression Analysis<br>Multilevel regression analysis (Linear and logistic regression models) | Odds ratios, 95% CIs, p-values | NR | Individual, family, community | NR |
| **Austin et al. (2019)** | Advanced Regression Analysis<br>1. Multinomial logistic regression<br>2. Growth Mixture Modelling | 1. Trajectory class probabilities<br>2. Lo-Mendell-Rubin Adjusted Likelihood Ratio test, P-value | Yes | Individual, family | NR |
| **Austin et al. (2019)** | Survival Analysis<br>1. Kaplan-Meier method<br>2. Cox (proportional hazards) regression. | 1. Cumulative incidence proportion<br>2. 95% CI, Hazard ratios, p-values | NR | Individual, family | NR |
| **Hansson et al. (2020)** | Regression Analysis<br>Multiple Regression Analysis | Standard errors, t-statistic, p-values, 95% CI | Yes | Individual, family | NR |
| **Abajobir et al. (2017)** | Regression Analysis<br>Logistic Regression | Prevalence, Odds ratios, p-values, 95% CI | NR | Individual, family | NR |

*(Continued)*

**Table 4.** (Continued)

| Author | Domain & Analysis Procedure | Statistical parameters | Assumption test | Independent Variables | Mediation and Moderating Variables |
|---|---|---|---|---|---|
| **Abajobir et al. (2017)** | Regression Analysis Logistic Regression | Prevalence, Odds ratios, p-values, 95% CI. | NR | Individual, family | NR |
| **Abajobir et al. (2017)** | Regression Analysis Logistic Regression | Prevalence, Odds ratios, p-values, 95% CI. | NR | Individual, family, community | NR |
| **Abajobir et al. (2017)** | Regression Analysis Logistic Regression | Prevalence, Odds ratios, p-values, 95% CI. | NR | Individual, family | Mediating: Gender |
| **Abajobir et al. (2017)** | Regression Analysis Logistic Regression | Prevalence, Odds ratios, p-values, 95% CI. | NR | Individual, family | Mediating: Gender |
| **Strathean et al. (2009)** | Regression Analysis Logistic Regression | Prevalence, Odds ratios, p-values, 95% CI. | NR | Individual, family | NR |
| **Mills et al. (2013)** | Regression Analysis Multiple Regression Analysis | Mean differences in internalizing and externalizing scores, regression coefficients, 95% CI | NR | Individual, family | NR |
| **Mills et al. (2016)** | Regression Analysis Logistic Regression | Odds ratios, 95% CI, p-values | NR | Individual, family | NR |
| **Mills et al. (2014)** | Regression Analysis Logistic Regression | Odds ratios, 95% CI, p-values | NR | Individual, family | Mediating: Smoking & alcohol use at 14 year follow-up |
| **Mills et al. (2019)** | Regression Analysis 1. Multiple Regression Analysis 2. Logistic Regression | 1. Frequencies, percentages, mean scores, standard deviation, Population Attributable Risk (PAR%), Unstandardised regression coefficients, 95% CI, p-values; 2. Odds ratio, 95% CI, p-values | NR | Individual, family | NR |
| **Mills et al. (2017)** | Regression Analysis Logistic Regression | Odds ratio, 95% CI, p-values | NR | Individual, family | NR |
| **Parrish et al. (2011)** | Regression Analysis Logistic Regression | Beta coefficient, standard errors, Wald F statistic, p-values, 95% CI, Odds ratio | NR | Individual, family | Mediating Public aid, race |
| **Raghavan et al. (2012)** | Regression Analysis 1. Logistic Regression 2. Generalized linear model (GLM) | 1. Odds ratios, 95% CI, p-value; 2. GLM coefficients, 95% CI, p-value | NR | Individual | NR |

Notes

CPS Child Protective Services

CI Confidence Interval

LTFC Long Term Foster care

DVSA Domestic violence and sexual assault

N/A Not Applicable

NR Not Reported

PPVT Peabody Picture Vocabulary Test

## Discussion

This systematic review sought to describe the study designs and statistical methods used when administrative data is integrated with longitudinal data in child protection settings and make recommendations about approaches to improve the quality of reporting of research findings, thereby minimising risk of bias and other limitations. There has been a steady growth in the number of studies which use administrative data integrated with longitudinal data in child protection settings since 2000. A total of 30 studies were identified that integrated these data to determine outcomes in the areas of child maltreatment, mental health, drug and alcohol abuse

Table 5. Study description.

| Author | Sample Size | Confounders | Outcome |
|---|---|---|---|
| **Egulend et al, (2009)** | OHC (1,072);<br>In-home care (1,457);<br>Non- Child Protection Contacts (71,321) | All Children, Children in out-of-home care, In-home care children, non-welfare children, number of siblings, Danish born children, Mother's age, teenage mothers, single mothers, mother's education, mother's employment status, mother/ father died, mother/ father with a psychiatric illness, mother/ father substance abuse problem, mother/ father previously convicted, mother/ father in care as children | Clinical diagnosis of psychiatric illnesses |
| **Hansson et al. (2018)** | Non-OHC (40,107);<br>OHC (1,482) | Gender, migration, parents' education, OHC vs Non-OHC, relocations | 1. Cognitive Ability Test Level;<br>2. Special Needs Education |
| **Kisely et al. (2019)** | Smoking status (3,758);<br>Nicotine use dependence (2,548);<br>Propensity Analysis (7,223) | Gender of the child, parental race, maternal age, mother's relationship status, family income at study entry (first prenatal visit), maternal smoking, and maternal education at study entry, childhood maltreatment | 1. Cigarette smoking;<br>2. Any cigarette use;<br>3. Long-term cigarette use;<br>4: CIDI-Auto (12-month Nicotine use disorder) |
| **Kisely et al. (2018)** | 1. YASR(3,725);<br>2. CIDI-Auto (2,508);<br>3. CES-D (3,778) | Gender of the child; parental ethnicity; maternal age; mother's relationship status; family income at the time of study entry (first prenatal visit) and maternal education status at study entry, overall child maltreatment, emotional, physical, sexual abuse, neglect. | 1. YASR (Internalising & Externalising);<br>2. CIDI, DSM-IV (Depression, Anxiety, PTSD)<br>3. CES-D |
| **Kisely et al. (2019)** | 1. Alcohol use in the last month (3,762);<br>2. Alcohol use disorder (2,531) | First prenatal visit (Race, maternal age, mother's education, marital status and family income) and at 21-year follow up (employment, marital status, educational level and residence in a problem area), childhood maltreatment | 1. Alcohol use in the last month;<br>2. CIDI DSM-IV Alcohol use disorder |
| **Olsen et al. (2018)** | 1. OHC (107);<br>2. Non-OHC (3,805) | Gender, birth weight, ethnicity, citizenship, psychiatric diagnosis, bullying, family type, mother's educational level, father's educational level, mother's disposable income, father's disposable income. | 1. Self perceived academic ability at age 15 years;<br>2. School change in lower secondary school |
| **Parrish et al. (2016)** | Total (2,389) | Self-reported IPV, race, maternal education, maternal smoking, maternal alcohol use, poverty, parents marital status, prenatal care, maternal age | Maltreatment report to Child Protective Services |
| **Parrish et al. (2017)** | Total (1,235) | Birth paid by Tricare (military families), sex of the child, maternal education at child's birth, marital status at birth, maternal alcohol use during pregnancy, maternal smoking during pregnancy, maternal race, birth defect, mother or child on Medicaid at birth, fathers name listed on birth certificate, maternal age at birth, multi-agency maltreatment report, mother reported being divorced/separated 12 months before pregnancy, mother reported moving 12 months before pregnancy, mother reported losing a job 12 months before pregnancy, mother reported partner/ husband losing a job 12 months before pregnancy | Censorship;<br>Multi-source report of maltreatment |
| **Raghavan et al. (2017)** | LTFC (1,569);<br>CPS (8,917) | Age, gender, race/ethnicity, Insurance type, primary care case management, urban/rural location, health condition, health care access | Ascertainment of foster care status |
| **Sidebotham et al. (2000)** | 1.Registered children (139);<br>2. Children investigated but not registered (190);<br>3. Children neither investigated nor registered (13, 927) | 1. Time period (8, 18, 21, 30 33 months);<br>2. Registered children; children investigated but not registered; children neither investigated nor registered | 1. Rates of child protection registrations;<br>2. Proportion of child abuse investigations and registrations;<br>3. Parental reporting of child abuse |
| **Sidebotham et al. (2003)** | 1. Registered children (115)<br>2. Non-registered children (14,105) | Low birthweight, unintended pregnancy, hospital admissions, developmental concerns, reported positive attributes, feeding difficulties, temper tantrums, parental concerns about the child's development, and not seeing the child in a positive light. | Child protection registration prior to 6 years of age |
| **Sidebotham et al. (2006)** | Registered children (115);<br>Investigated children (178);<br>Neither registered nor investigated (13, 963) | Parental ontogenic background (Young parent, low educational achievement, psychiatric history, history of childhood abuse (any); Exosystem (socio-demographic) variables (Any indicator of poverty, Mother employed, Poor social network. Microsystem (family) variables (high parity, single mother, reported domestic violence, reordered family); Child variables (Unintended pregnancy, Low birthweight, Few positive attributes reported | 1. Children registered for maltreatment;<br>2. Children investigated for maltreatment |
| **Sidebotham et al. (2002)** | Registered children (85);<br>Non-registered children (13, 089) | Maternal employment, mobility (house moves), social network score. | 1. Child Abuse registrations<br>2. Child maltreatment |

(*Continued*)

**Table 5.** (Continued)

| Author | Sample Size | Confounders | Outcome |
|---|---|---|---|
| **Teyhan et al. (2019)** | No CLA or CIN (9,432); CIN(64); CLA (49) | Social care status, Age, sex, socio-economic position, maternal age, highest educational qualification; financial difficulties; housing tenure; partner status; smoking; alcohol intake; social support; and depressive symptoms | 1. Educational attainment; 2. Persistent absence from school; 3. Special educational needs (SEN) status; 4. School Mobility |
| **Austin et al. (2019)** | AN(1,253); NN (2,094) | Maternal age and education at childbirth, preconception and prenatal substance use, and experiences of emotional, traumatic, partner, and financial stress in the 12 months prior to childbirth | Longitudinal trajectory classes of CPS contact |
| **Austin et al. (2019)** | Total (3,549) | Maternal race, maternal age, maternal education, maternal marital status, residence at childbirth, number of living children, maternal history of pregnancy terminations, pregnancy intendedness, timing of prenatal care, number of stressful live events, maternal experience of intimate partner violence (IPV), maternal alcohol use, maternal smoking during pregnancy, maternal marijuana use, socioeconomic status, infant sex, infant birth defects | Age at first CP contact |
| **Hansson et al. (2020)** | OHC (1,099); Non OHC(30, 936) | Gender, migration, parents' education, school relocations, Cognsum | Academic achievement |
| **Abajobir et al. (2017)** | Total (3,752) | Youth gender, ADHD at 5 year, alcohol use, smoking, aggressive behaviour (at 14 years), receiving benefits, educational levels, marital status, residential problem area at 21 years, familial income over the first 5 years, chronic stress over first 6 months, and maternal reports of violence in homes at 14 years, any abuse, sexual abuse, physical abuse, emotional abuse, neglect | 1. Auditory hallucinations 2. Visual hallucinations 3. Peter's Delusional Inventory (PDI) 4. DSM-IV Psychosis |
| **Abajobir et al. (2017)** | Total (3,730) | Child maltreatment, maternal age at first clinic visit, family income at first clinic visit, gender at birth, educational status, receipt of social security benefits and depressive symptoms at 21-year follow-up | Quality of Life Index Score |
| **Abajobir et al. (2017)** | Total (3,322) | Substantiated child maltreatment, sex at birth, receipt of social security benefits, educational level, marital status and residential problem area at 21-year, aggressive child behaviour, maternal poverty level, maternal marital stability, maternal stress, maternal negative life events, family violence | Intimate partner violence victimization |
| **Abajobir et al. (2017)** | Total (2,526) | Any maltreatment, sexual abuse, physical abuse, neglect, emotional abuse, age at substantiation, frequency of substantiation, maternal age at pregnancy, maternal prenatal and postnatal cigarette smoking, family poverty, educational level, marital status, gender at birth | Cannabis abuse, dependence, early age of onset of cannabis abuse and dependence |
| **Abajobir et al. (2017)** | Total (3,750) | Any maltreatment, sexual abuse, physical abuse, neglect, emotional abuse, receiving social security benefits, educational level, marital status at 21 years and paternal or maternal race at pregnancy, maternal alcohol use at 3–6 months and chronic depressive symptoms | Injecting drug use |
| **Strathean et al. (2009)** | Total (5,890) | Maternal prenatal demographic factors (age, marital status, education, race, employment); prenatal behaviours/attitudes (cigarette consumption and binge drinking during pregnancy, anxiety and pregnancy ambivalence); infant factors (birth weight and gender), and 6 month postpartum maternal behaviours and attitudes (mother-infant separation, employment, maternal stimulation/teaching of baby, maternal attitude of caregiving and postpartum depression). Models: 1. Breastfeeding duration, 2. Single vs. multiple episodes of maltreatment, 3. Exclude previously enrolled children, 4. Only children in Queensland at 14 years of age | Substantiated maternal child maltreatment |
| **Mills et al. (2013)** | Total (5,098) | Notified and substantiated maltreatment, type of maltreatment (exclusive; hierarchical scheme), gender, race, During pregnancy (maternal age, marital status, maternal education) family income prior to birth | Internalizing and externalizing scales of the Youth Self Report (YSR) |
| **Mills et al. (2016)** | Major depressive disorder (2, 304); Anxiety disorder (2,298); PTSD(2,292) | Self-reported CSA, Agency-notified CSA, Agency-substantiated CSA, gender, parental race, maternal age, maternal relationship status, family income, and maternal education | Major depressive disorder; Anxiety disorder; PTSD |

(*Continued*)

**Table 5.** (Continued)

| Author | Sample Size | Confounders | Outcome |
|---|---|---|---|
| **Mills et al. (2014)** | Any alcohol use (5,153); Any smoking (5,154) | Maltreatment notification, type of maltreatment, Family income, maternal alcohol use and maternal smoking (14y follow-up); maternal education and marital status (prenatal); and race, age, and gender. | Alcohol use; smoking |
| **Mills et al. (2019)** | 1. Peabody Vocabulary Test (2,150); 2. Failure to complete high school (3,750); 3. Failure to be employed or in education (3,739) | Notified maltreatment, substantiated maltreatment, age, sex, race, family income, maternal education, birthweight $z$ score, neonatal intensive care admission, maternal tobacco and alcohol use in pregnancy, breast feeding | 1. Peabody picture vocabulary test 2. Failure to complete high school 3. Failure to be employed or education at 21 years |
| **Mills et al. (2017)** | Total (3,778) | Age, gender, race, family income, and maternal age, education, marital status, alcohol use, smoking, anxiety and depression, maltreatment type, additional adjustment for youth smoking and alcohol use at 14-year follow-up, youth internalizing and externalizing scale | Cannabis use/ dependence |
| **Parrish et al. (2011)** | Total Population (28,592); PSR (3,271) | Maternal age and education, DVSA (maternal physical abuse and forced sexual activities), Maternal tobacco use, Maternal marital status, Substance abuse, living children, medically vulnerable, public aid, risk group category | PSR to child protective services |
| **Raghavan et al. (2012)** | Total (5,652) | Child age, gender, race/ ethnicity, rural/urban location, insurance type, placement status, health status, CBCL score, maltreatment type | 1. Annual probability of having any medication expenditures 2. Expenditures per child per year |

Notes

**AN** Alaska Native

**CI** Confidence Interval

**CIDI** Composite International Diagnostic Interview

**CES-D** Centre for Epidemiological Studies–Depression Scale

**CLA** Children Looked After

**CIN** Children In Need

**CP** Child Protection

**CPS** Child Protective Services

**DSM-IV** Diagnostic and Statistical Manual of Mental Disorders, 4[th] edition

**DVSA** Domestic violence and sexual assault

**IPV** Intimate Partner Violence

**LTFC** Long Term Foster Care

**N/A** Not Applicable

**NR** Not Reported

**NN** Non-Native

**OHC** Out-of-home care

**PPVT** Peabody Picture Vocabulary Test

**PSR** Protective Services Report

**PTSD** Post-Traumatic Stress Disorder

**SDQ** Strength and Difficulties Questionnaire

**YASR** Young Adult Self Report

and education. Since the focus of the review was on studies in child protection settings, the main administrative data reported was child protection data.

While most studies had multiple data collection points, the median number of waves reported for the longitudinal studies was two. The findings from this review can be grouped under three themes: i) quality of reporting on data linkage procedures; ii) biases reported; and iii) statistical methods used. Though some systematic reviews have been conducted on administrative data alone or longitudinal data alone in child protection or other settings [26, 110,

**Table 6. Quality appraisal of included studies.**

| Study | Qualsyst (KMET) | | GUILD and RECORD | |
|---|---|---|---|---|
| | Score (%) | Methodology Quality | Score (%) | Methodology Quality |
| Egulend et al. (2009) | 50% | Adequate | 24% | Poor |
| Hansson et al. (2018) | 68% | Good | 10% | Poor |
| Kisely et al. (2019) | 91% | Strong | 26% | Poor |
| Kisely et al. (2018) | 91% | Strong | 22% | Poor |
| Kisely et al. (2019) | 91% | Strong | 22% | Poor |
| Olsen et al. (2018) | 86% | Strong | 21% | Poor |
| Parrish et al. (2016) | 82% | Strong | 33% | Poor |
| Parrish et al. (2017) | 86% | Strong | 79% | Good |
| Raghavan et al. (2017) | 86% | Strong | 33% | Poor |
| Sidebotham et al. (2000) | 60% | Good | 10% | Poor |
| Sidebotham et al. (2003) | 80% | Strong | 16% | Poor |
| Sidebotham et al. (2006) | 91% | Strong | 29% | Poor |
| Sidebotham et al. (2002) | 91% | Strong | 16% | Poor |
| Teyhan et al. (2019) | 91% | Strong | 28% | Poor |
| Austin et al. (2019) | 86% | Strong | 72% | Good |
| Austin et al. (2018) | 95% | Strong | 71% | Good |
| Hansson et al. (2020) | 73% | Good | 9% | Poor |
| Abajobir et al. (2017) | 95% | Strong | 22% | Poor |
| Abajobir et al. (2017) | 95% | Strong | 26% | Poor |
| Abajobir et al. (2016) | 95% | Strong | 26% | Poor |
| Abajobir et al. (2016) | 95% | Strong | 26% | Poor |
| Abajobir et al. (2017) | 91% | Strong | 29% | Poor |
| Strathean et al. (2009) | 95% | Strong | 47% | Poor |
| Mills et al. (2013) | 95% | Strong | 22% | Poor |
| Mills et al. (2016) | 95% | Strong | 21% | Poor |
| Mills et al. (2014) | 95% | Strong | 21% | Poor |
| Mills et al. (2019) | 91% | Strong | 16% | Poor |
| Mills et al. (2017) | 100% | Strong | 16% | Poor |
| Parrish et al. (2011) | 95% | Strong | 19% | Poor |
| Raghavan et al. (2012) | 100% | Strong | 45% | Poor |
| **Median** | **91%** | Strong | **23%** | Poor |

111], this is the first systematic review of studies utilising administrative data integrated with longitudinal data in child protection settings.

## Quality of reporting on data linkage procedures

Overall, the quality of all studies was strong (Qualsyst median score = 93%), but most of the studies rated poorly on the reporting of data linkage methods (GUILD and RECORD median score = 23%). Only three of the 30 studies [55, 92, 99] described the data linkage procedures in sufficient detail. This is of concern, as a small amount of data linkage errors may lead to significant bias and inconsistencies in estimating parameters of a statistical model. As described in the GUILD [7], researchers utilising linked data should take account of biases inherent in the data linkage process and account for such biases in the analysis. The GUILD guidelines recommend following three key steps when reporting analyses using linked data: i) describing the population included in the data set (i.e., how the data were generated, processed and quality

controlled); ii) describing the data linkage processes; and iii) describing the quality of data linkage, including accounting for linkage error. Similar reporting items are recommended in the RECORD statement [79].

Harron, Dibben [38] supports the notion of accounting for linkage errors as recommended by GUILD and RECORD, but states that it may be difficult for researchers to determine the quality of linked data since researchers may not have access to identifiable data. The authors therefore recommend conducting the following three methods to evaluate data linkage quality and identify potential sources of bias: i) post-linkage validation, ii) sensitivity analyses, and iii) comparison of characteristics of linked and unlinked records.

Most authors did not report sufficiently on the population included in the data set and how the data were generated and quality controlled. Most authors provided descriptions of the population in the source data and how the data were collected, but no information was reported on how the data were updated, processed and quality controls. Only a few authors explained how data were cleaned, including standardisation of missing data and treatment of special characters [55, 92, 99], and how manual linkages were conducted by reporting on data mismatches and duplicate cases [86].

The second GUILD step, which focusses on data linkage processes, was described in sufficient detail by the same authors [55, 92, 99] by reporting on how linkage rates were calculated and how probability match scores were used for weighting. Benchimol, Smeeth [79] state that the methods of linkage and methods of linkage quality evaluation should be reported by authors, though this information may not be provided by the data linkage unit. Furthermore, information on disclosure controls to reduce the re-identification of individuals from linked data was not reported in any of the studies. However, the majority (80%) of studies reported the method of data linkage (deterministic or probabilistic, or both), including reporting the unique ID that was used as the variable for deterministic linkage.

The last GUILD step involves analysis of linked data which takes linkage error into account. While the quality of data linkage can be determined prior and during data linkage, this step allows researchers to report on linkage error post data linkage. The analysts who conduct data linkage should provide researchers with reports of the data linkage process, including estimates of false and missed matches, so that there is transparency. If there are linkage errors, analysts can determine methods or procedures to correct for this before conducting any analysis, while acknowledging this may not always be possible [7]. Analysts could identify linkage errors by analysing differences or similarities between linked and unlinked data [112], though this method may introduce additional bias caused by missing records [10]. A simulation exercise developed by Parrish, Shanahan [55] enables post-estimation of linkage errors. The inclusion of linkage errors into research analyses is an evolving and relatively new area of methodological research. Some methods that have been developed by researchers model simple linkage errors derived from one-to-one matches rather than the more complex many-to-many or many-to-spine match scenarios that exist in modern day production linkage systems. [112, 113]

## Biases reported

In longitudinal studies there is commonly missing data for various reasons, such as non-availability of data from specific variables or missing data due to participant attrition. Missing data may result in loss of statistical power, bias in estimation of parameters, and diminish the representativeness of samples in a study [114]. Almost all studies described missing data and a few conducted some analysis to correct for missing data. Biases may occur due to certain population groups being over-represented, for instance Aboriginal children are over-represented in

child-protection or out-of-home care systems compared with other young people in Australia. Systematic bias may occur as a result of Aboriginal young people being more often reported and therefore at increased contact with child protection services. Some studies reported over-representation of children in OHC among those with missing school grades and this was corrected by replacing the missing grades with estimated grades (MAR) [89, 90]. If the missing data were not accounted for in the analysis this could have resulted in over or under-estimation of outcomes among the OHC group.

This review shows some variability in the reporting and analysis of missing data. A review conducted by Karahalios, Baglietto [43] highlighted that there is generally inconsistent reporting of missing data in cohort studies and methods employed to handle missing data in some studies may be inappropriate. While weighting was described as one technique to account for missing data, this method has limitations. For example, standard errors of estimates, such as means and proportions, are larger than they would be if the data were not weighted [115].

Listwise deletion as a method of handling missing data also has limitations as it requires data to be MCAR [116]. While some studies in this review applied this method it may not be appropriate, particularly if the missing values occur among populations with certain characteristics, such as those lost to follow up who were mostly disadvantaged or are hard to reach. In addition, listwise deletion results in a reduced sample size (and ultimately loss of statistical power), which is a concern particularly among young people with child protection contact where smaller sample sizes are reported compared to comparison groups in the general population.

## Statistical methods

Most studies reported using logistic regression as a method of analysing the factors associated with reported outcomes. While this method was appropriate to determine the impact of reported outcomes with a binary scale, controlling for multiple confounders, more sophisticated methods of analysis were expected, particularly where mediating or moderating effects of some variables were required. One of the limitations in the reporting of logistic regression analysis was lack of descriptions on why this method was chosen in relation to fulfilling the assumption that there is a linear relationship between the logit of the outcome and each predictor variables. Likewise, with multiple regression methods the assumption of linearity has to be satisfied; this was not often described where linear regression methods were used.

Survival analysis methods were well described and utilised where there were more than two pre-specified time points and these included the Nelson-Aalen Estimation method [55], the Kaplan-Meier method, and the Cox regression method [99]. Three studies described more advanced methods of analysis which are Multinomial logistic regression model using Vermunt's three step Latent Class Analysis Approach, Growth mixture modelling and Generalised Linear Model [92, 108]. Sensitivity analysis was conducted particularly when definitions of child maltreatment were altered to either include substantiated maltreatment or reported allegations. Conducting sensitivity analysis prior to data modelling may not be necessary since sensitivity analysis is usually done after a statistical model has been estimated and the results interpreted [117].

The statistical methods applied to most of the included studies lack the sophistication expected of longitudinal studies with certain covariance structures. The methods used fail to take into account random or systematic error which may be inherent to the measurable observed variables [118]. Failure to account for such errors in the analysis may lead to under or over estimation of the true values of the measured outcomes. This limitation can only be overcome by using techniques such as structural equation modelling (SEM) that estimates

latent variables which are not directly observed and which provide a closer estimation to measurement error for each observed variable [119]. Only one study used multi-level modelling; an analytical approach with similar benefits to SEM [45]. These methods were not explored in other studies as a technique for analysing longitudinal data where outcomes are studied over time (i.e., involving multiple data collection points) or accounting for the correlation of individual responses over time. This is surprising given the usefulness of these methods when analysing participants with varying lengths of follow-up due to death and MAR outcomes [120].

SEM also allows the estimation of the indirect effect of mediating variables on outcomes of interest [121, 122]. Seven studies [21, 47, 58, 89, 93, 97, 102] reported the role of mediating variables, without reporting on the indirect effects that these variables have on outcomes. Most authors reported several logistic regression models per study, whereas SEM is able to model multiple regression equations simultaneously, and hence provides a flexible framework for testing a range of possible relationships between the variables in the model, including mediating effects and possible latent confounding variables [123, 124].

Logistic regression analysis and multiple linear regression analysis assume a direct pathway analysis and, therefore, fail to take into account mediating factors which may have an indirect effect on the outcomes of interest [123]. More recently, Bayesian methods have been proposed as important complementary approaches for testing for mediation and computing the value of the mediation effect (often referred to as Bayesian Mediation Analysis) [125, 126]. Literature has determined that Bayesian methods of analysis are better suited to analyse data with small sample sizes as compared to frequentist methods, though it is important that the prior distribution is correctly specified to avoid obtaining less accurate estimates [117, 127].

## Strengths and limitations

This review has several strengths. The systematic search used a comprehensive range of databases including directed search strategies from linked child protection data and longitudinal study websites and manual scrutiny of reference lists were conducted. The integrity of the review process was maintained through quality control procedures including independent assessment of the included and excluded studies. However, the review was limited to peer reviewed studies published in English only, thus limiting the ability to review unpublished studies and studies from non-English speaking countries. Future reviews should consider targeted searches that may uncover literature from other geographic regions such as Asia, Africa and South and Central America.

## Recommendations for future research

Overall, the quality of studies was good but the reporting of data linkage procedures was poor. It is important that in future, researchers should conduct adequate data preparation consisting of checking for errors and missing data and ways to address these. Additionally, the generalisability of the findings on the reported studies may be questionable as the reporting omitted important aspects of mediation analysis and ways to overcome bias due to small sample sizes.

The review has shown that it is important that researchers follow the guidelines recommended by the GUILD and RECORD statements to report the quality of data linkage so that there is transparency in the reporting process. While some data linkage communities have recognised the need to improve on their reporting of linkage quality to researchers it remains apparent that there should be improved communication and engagement between researchers and the data linkage units so that the reporting of linkage quality can be provided more routinely and consistently [128]. The poor or lack of transparency in reporting data linkage processes, such as reports on linkage errors, may under or overestimate the quality of studies

reported, particularly among the hard to reach populations as exemplified in these studies. The more vulnerable or hard to reach populations are often missed or miss matches, resulting in reduced sample size and loss of statistical power [10, 129].

In addition, our review has also shown that there was lack of reporting or referencing of validated data quality assessments conducted for administrative data. In the context of transparency, accuracy, and reliability of measurement from administrative data sources, it is important to reference validated appraisal tools. Additionally, due to variability in quality criteria for child protection administrative data sets, we recommend that future researchers implement a data quality framework [130, 131]. With the growing use of administrative data it is necessary that data quality indicators are operationalised and reported in studies. For example, leaders in the use of linked administrative data at the Manitoba Centre of Health Policy have identified 5 dimensions of data quality: accuracy, internal validity, external validity, timeliness, and interoperability.

These dimensions of data quality can serve as an important starting point for future reporting of administrative data. However, determining if these dimensions are comprehensive, what exact criteria should be used for each dimension, and the operationalisation of those dimensions into measurable data quality criteria remains elusive. As such, there is need to conduct a Delphi Study [132, 133] among leading experts in the field of administrative data, to establish consensus on the use of these data quality indicators to either be integrated into tools such as the GUILD [7] and RECORD [79] guidelines, or to develop a new comprehensive data quality appraisal tool.

Reporting of missing data may be done by following some recommended guidelines such as the STROBE [134] and RECORD [79] guidelines. According to these guidelines, the number of individuals used for analysis at each stage of the study should be reported followed by reasons for non-participation or non-response. When it comes to handling missing data, simple to more complex analytical methods should be applied and the method used should take into account the mechanism for missingness [114]. If a wrong technique is applied, this may lead to biased inferences [135].

If data is MCAR, listwise deletion can be conducted because the reason for missing data is unrelated to the data itself. Pairwise deletion can be used as an alternative to listwise deletion since it preserves more information than listwise deletion [114]. While if data is MAR, analysis of complete records only may be invalid and thus techniques such as multiple imputation and likelihood based methods should be applied, though if not carried out appropriately, this could lead to biased estimates. If the reason for missing data depends on the missing values (NMAR), it is important to account for this by modelling the missing data and thus avoid getting parameters with biased estimates.

Basic regression methods of analysis were reported in most studies. More advanced statistical techniques, such as SEM and Bayesian, should be incorporated in analysis of cohort studies, particularly where small sample sizes are involved and where there are multiple data collection time points and multiple covariates. Multilevel structural equation modelling (ML-SEM) combines the advantages of multi-level modelling and structural equation modelling and further enables researchers to scrutinize complex relationships between latent variables at different levels [136].

## Conclusions

Studies utilising administrative data integrated with longitudinal data in child protection settings were homogenous in nature. Most were birth cohort studies that were integrated with child protection data. There was poor reporting of data linkage processes, whereby only three

studies (10%) reported the data linkage process in sufficient detail. A few techniques to account for missing data were reported, but generally lacked sufficient analytical details. The main statistical method of analysis reported in most studies were regression analysis which fail to take into account mediating factors which may have an indirect effect on the outcomes of interest. Furthermore, there was lack of utilisation of multi-level analysis as would have been expected in longitudinal studies reported where an individual's responses over time are correlated with each other. While a few studies (10%) reported advanced statistical analysis methods, there is an opportunity to implement other advanced techniques in future studies where small samples are involved. Additionally, the methods should account for measurement and linkage errors and missing data due to attrition. The review emphasises the need for more effort to be channelled towards improvements in reporting of data linkage processes through following recommended and standardised data linkage processes, which can be achieved through greater co-ordination among data providers and researchers.

## Supporting information

**S1 Table. PRISMA checklist.**
(DOCX)

**S2 Table. Search strategy from all databases.**
(DOCX)

**S3 Table. Data preparation methods.**
(DOCX)

**S1 File.**
(PDF)

## Author Contributions

**Conceptualization:** Fadzai Chikwava, Reinie Cordier, Anna Ferrante, Melissa O'Donnell.

**Data curation:** Fadzai Chikwava.

**Formal analysis:** Fadzai Chikwava, Reinie Cordier, Lauren Parsons.

**Investigation:** Fadzai Chikwava, Reinie Cordier, Renée Speyer, Lauren Parsons.

**Methodology:** Fadzai Chikwava, Reinie Cordier, Anna Ferrante, Melissa O'Donnell, Lauren Parsons.

**Supervision:** Reinie Cordier, Anna Ferrante, Melissa O'Donnell.

**Writing – original draft:** Fadzai Chikwava.

**Writing – review & editing:** Reinie Cordier, Anna Ferrante, Melissa O'Donnell, Renée Speyer, Lauren Parsons.

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
