## [Decision Letter · Decision Letter 0]

5 Jan 2021

PONE-D-20-29134

Research using population-based administration data integrated with longitudinal data in child protection settings: A systematic review

PLOS ONE

Dear Dr. Chikwava,

Thank you for submitting your manuscript to PLOS ONE. After careful consideration, we feel that it has merit but does not fully meet PLOS ONE’s publication criteria as it currently stands. Therefore, we invite you to submit a revised version of the manuscript that addresses the points raised during the review process.

Your review was assessed by two independent reviewers and was found to be of interest to the readership and meeting the required publication criteria. The review was deemed very well by reviewers. However, minor revisions were suggested. Please address them as soon as possible. 

We look forward to receiving your revised manuscript.

Kind regards,

Abraham Salinas-Miranda, MD, PhD

Academic Editor

PLOS ONE

Journal Requirements:

Additional Editor Comments:

Dear authors, your review was assessed by two independent reviewers and was found to be of interest to the readership and meeting the publication criteria. The review was deemed very well by reviewers and just minor revisions were suggested.

Reviewers' comments:

Reviewer's Responses to Questions

**Comments to the Author**

1. Is the manuscript technically sound, and do the data support the conclusions?

Reviewer #1: Yes

Reviewer #2: Yes

2. Has the statistical analysis been performed appropriately and rigorously? 

Reviewer #1: Yes

Reviewer #2: Yes

3. Have the authors made all data underlying the findings in their manuscript fully available?

Reviewer #1: Yes

Reviewer #2: Yes

4. Is the manuscript presented in an intelligible fashion and written in standard English?

Reviewer #1: Yes

Reviewer #2: Yes

5. Review Comments to the Author

Reviewer #1: I found the review inspiring and enjoyed reading it.

One of the main results (lack of information on data linkage process and quality) clearly is important for future studies.

What seems to be missing is an analysis and discussion on quality criteria for adminstrative data sets as we and others have shown that there are huge differences (e.g. reliability of substantiation process for child maltreatment notifications)

Reviewer #2: The Research article is a systematic review which aims at describing the study designs and statistical methods performed when integrating longitudinal and administrative data in child protection settings. The systematic review seems to be novel as it is reported as the first systematic review of studies assessing both administrative data and with longitudinal data in child protection settings. The methods are sound and rigorous, guided by the PRISMA, conducted in a variety of database to ensure that all eligible peer reviewed articles are captured. A rigorous screening and study selection were performed as well as the use of quality assessment tools with the majority of eligible studies ranked as high-quality studies (Qualsyst median score=93%). The eligible studies statistical methods, bias and outcomes were investigated and succinctly reported. Bias was reported on the most common causes including attrition, missing data and selection bias. The conclusion is consistent with and aligned with the findings.

For future studies, the researcher can expand his/her inclusion criteria and geographic scope, assessing peer reviewed and non-peer reviewed publications, published in languages other than English and regions including Africa and Asia.

6. PLOS authors have the option to publish the peer review history of their article (what does this mean?). If published, this will include your full peer review and any attached files.

Reviewer #1: No

Reviewer #2: **Yes: **Marlene Joannie Bewa

---

## [Author Response · Author response to Decision Letter 0]

24 Jan 2021

To PLOS One

Reviewers and Academic Editors 

Dear Sir/ Madam

Re: Response to reviewers

We would like to thank the Reviewers and Editors for the generous comments on the manuscript and have edited the manuscript to address their concerns. The responses are as follows:

Comments from Reviewer #1: “I found the review inspiring and enjoyed reading it.

One of the main results (lack of information on data linkage process and quality) clearly is important for future studies. What seems to be missing is an analysis and discussion on quality criteria for administrative data sets as we and others have shown that there are huge differences (e.g. reliability of substantiation process for child maltreatment notifications)”

Response: This is an important point raised by the reviewer and we have added in an additional recommendation in the manuscript to address this issue with four additional references:

Additional text in manuscript: (Lines 797-816)

“In addition, our review has also shown that there was lack of reporting or referencing of validated data quality assessments conducted for administrative data. In the context of transparency, accuracy, and reliability of measurement from administrative data sources, it is important to reference validated appraisal tools. Additionally, due to variability in quality criteria for child protection administrative data sets, we recommend that future researchers implement a data quality framework [130, 131]. With the growing use of administrative data it is necessary that data quality indicators are operationalised and reported in studies. For example, leaders in the use of linked administrative data at the Manitoba Centre of Health Policy have identified 5 dimensions of data quality: accuracy, internal validity, external validity, timeliness, and interoperability. 

These dimensions of data quality can serve as an important starting point for future reporting of administrative data. However, determining if these dimensions are comprehensive, what exact criteria should be used for each dimension, and the operationalisation of those dimensions into measurable data quality criteria remains elusive. As such, there is need to conduct a Delphi Study [132, 133] among leading experts in the field of administrative data, to establish consensus on the use of these data quality indicators to either be integrated into tools such as the GUILD [7] and RECORD [79] guidelines, or to develop a new comprehensive data quality appraisal tool.”

Additional References:

1. Laitila T, Wallgren A, Britt, Wallgren B. Quality Assessment of Administrative Data. 2011.

2. Smith M, Lix LM, Azimaee M, Enns JE, Orr J, Hong S, et al. Assessing the quality of administrative data for research: a framework from the Manitoba Centre for Health Policy. Journal of the American Medical Informatics Association : JAMIA. 2018;25(3):224-9. doi: http://dx.doi.org/10.1093/jamia/ocx078. PubMed PMID: 1988263660; 29025002.

3. Chia-Chien H, Sandford BA. The Delphi Technique: Making Sense of Consensus. Practical Assessment, Research & Evaluation. 2007;12:10. PubMed PMID: 2366822829.

4. Grisham T. The Delphi technique: a method for testing complex and multifaceted topics. International Journal of Managing Projects in Business. 2009;2(1):112-30. doi: http://dx.doi.org/10.1108/17538370910930545. PubMed PMID: 232630764.

Comments from Reviewer #2: “For future studies, the researcher can expand his/her inclusion criteria and geographic scope, assessing peer reviewed and non-peer reviewed publications, published in languages other than English and regions including Africa and Asia.”

Response: We have added an additional point in the limitation section: 

Additional text in manuscript: (Lines 770 - 773):

“Future reviews should consider targeted searches that may uncover literature from other geographic regions such as Asia, Africa and South and Central America.”

The authors hope that the Reviewers and Editors will be satisfied with the further amendments which we have made to the manuscript which we feel has strengthened the paper.

---

## [Decision Letter · Decision Letter 1]

11 Mar 2021

Research using population-based administration data integrated with longitudinal data in child protection settings: A systematic review

PONE-D-20-29134R1

Dear Dr. Chikwava,

We’re pleased to inform you that your manuscript has been judged scientifically suitable for publication and will be formally accepted for publication once it meets all outstanding technical requirements.

Kind regards,

Abraham Salinas-Miranda, MD, PhD

Academic Editor

PLOS ONE

Additional Editor Comments (optional):

The reviewers assessed the minor revisions as satisfactory and the recommendation for the decision is to "accept". Thank you for the prompt revisions.

Reviewers' comments:

Reviewer's Responses to Questions

**Comments to the Author**

1. If the authors have adequately addressed your comments raised in a previous round of review and you feel that this manuscript is now acceptable for publication, you may indicate that here to bypass the “Comments to the Author” section, enter your conflict of interest statement in the “Confidential to Editor” section, and submit your "Accept" recommendation.

Reviewer #1: All comments have been addressed

Reviewer #3: All comments have been addressed

2. Is the manuscript technically sound, and do the data support the conclusions?

Reviewer #1: Yes

Reviewer #3: Yes

3. Has the statistical analysis been performed appropriately and rigorously? 

Reviewer #1: N/A

Reviewer #3: Yes

4. Have the authors made all data underlying the findings in their manuscript fully available?

Reviewer #1: Yes

Reviewer #3: Yes

5. Is the manuscript presented in an intelligible fashion and written in standard English?

Reviewer #1: Yes

Reviewer #3: Yes

6. Review Comments to the Author

Reviewer #1: Thank you for your constuctive response to my remarks.

Reading the paper again reinforced the view that this is valuable contribution to child protection research.

Reviewer #3: I was asked to review the manuscript entitled "Research using population-based administration data integrated with longitudinal data in child protection settings: A systematic review" . The authors have addressed the limitations explicitly in this revision about a discussion for quality criteria for adminstrative data use and the need for greater geographical inclusion in future studies. This reviewer also agrees with their conclusion (based on findings) about the need to improve the reporting of data linkage processes for child protection research, which could benefit the field greatly.

7. PLOS authors have the option to publish the peer review history of their article (what does this mean?). If published, this will include your full peer review and any attached files.

Reviewer #1: No

Reviewer #3: **Yes: **Abraham A. Salinas-Miranda, MD, PhD. Harrell Center for the Study of Family Violence, University of South Florida College of Public Health, USA.

---

## [Editor Report · Acceptance letter]

15 Mar 2021

PONE-D-20-29134R1 

Research using population-based administration data integrated with longitudinal data in child protection settings: A systematic review 

Dear Dr. Chikwava:

I'm pleased to inform you that your manuscript has been deemed suitable for publication in PLOS ONE. Congratulations! Your manuscript is now with our production department. 

Kind regards, 

on behalf of

Dr. Abraham Salinas-Miranda 

Academic Editor

PLOS ONE